# On the Identification of Temporally Causal Representation with Instantaneous Dependence

**Zijian Li**[†•*] **Yifan Shen**[•*] **Kaitao Zheng**[‡] **Ruichu Cai**[‡] **Xiangchen Song**[†] **Mingming Gong**[*]
**Guangyi Chen**[†•] **Kun Zhang**[†•]
[†]Carnegie Mellon University, Pittsburgh PA, USA
[•]Mohamed bin Zayed University of Artificial Intelligence, Abu Dhabi, UAE
[‡]Guangdong University of Technology, Guangzhou, China
[*]The University of Melbourne

## Abstract

Temporally causal representation learning aims to identify the latent causal process from time series observations, but most methods require the assumption that the latent causal processes do not have instantaneous relations. Although some recent methods achieve identifiability in the instantaneous causality case, they require either interventions on the latent variables or grouping of the observations, which are in general difficult to obtain in real-world scenarios. To fill this gap, we propose an **ID**entification framework for instantane**O**us **L**atent dynamics (**IDOL**) by imposing a sparse influence constraint that the latent causal processes have sparse time-delayed and instantaneous relations. Specifically, we establish identifiability results of the latent causal process up to a Markov equivalence class based on sufficient variability and the sparse influence constraint by employing contextual information. We further explore under what conditions the identification can be extended to the causal graph. Based on these theoretical results, we incorporate a temporally variational inference architecture to estimate the latent variables and a gradient-based sparsity regularization to identify the latent causal process. Experimental results on simulation datasets illustrate that our method can identify the latent causal process. Furthermore, evaluations on human motion forecasting benchmarks indicate the effectiveness in real-world settings. Source code is available at https://github.com/DMIRLAB-Group/IDOL.

## 1 Introduction

Time series analysis (Zhang et al., 2023; Tang & Matteson, 2021; Li et al., 2023a; Wu et al., 2022; Luo & Wang, 2024), which has been found widespread applications across diverse fields such as weather (Bi et al., 2023; Wu et al., 2023b), finance (Tsay, 2005; Huynh et al., 2022), and human activity recognition (Yang et al., 2015; Kong & Fu, 2022), aims to capture the underlying patterns (Wang et al., 2019; Jin et al., 2022) behind the time series data. To achieve this, one solution is to estimate the latent causal processes (Tank et al., 2021; Li et al., 2023b; Zheng & Kleinberg, 2019). However, without further assumptions (Locatello et al., 2020), it is challenging to identify latent causal processes, i.e., ensure that the estimated latent causal process is correct.

Researchers have exploited Independent Component Analysis (ICA) (Hyvärinen & Oja, 2000; Comon, 1994; Hyvärinen, 2013; Zhang & Chan, 2007), where observations are generated from latent variables via a linear mixing function, to identify the latent causal process. To deal with nonlinear cases, different types of assumptions including sufficient changes (Yao et al., 2021; Khemakhem et al., 2020a; Xie et al., 2022b) and structural sparsity (Lachapelle et al., 2022; Zheng et al., 2022) are proposed to meet the independent change of latent variables. Specifically, several works leverage auxiliary variables to achieve strong identifiable results of latent variables (H"alv"a & Hyvarinen, 2020; Hälvä et al., 2021; Hyvarinen & Morioka, 2016; 2017; Khemakhem et al., 2020a; Hyvarinen

---
[*]These authors contributed equally to this work.

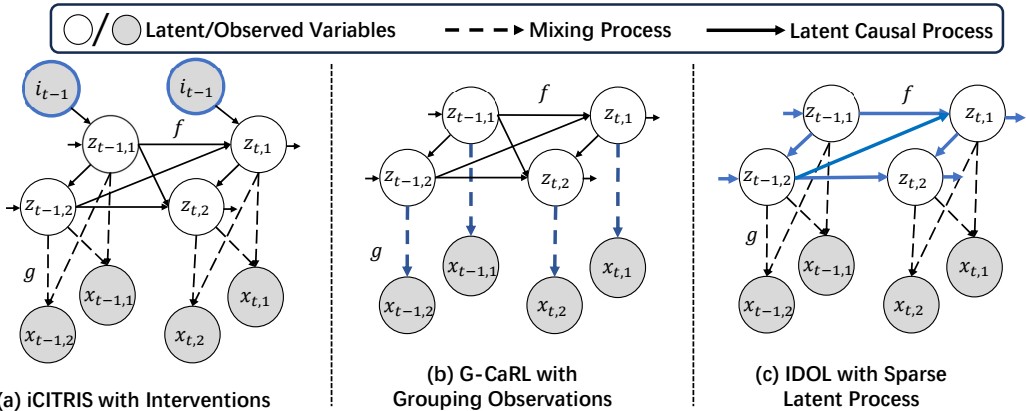

Figure 1: Three different data generation processes with time-delayed and instantaneous dependencies. (a) iCITRIS (Lippe et al., 2023) requires intervention variables $\mathbf{i}_t$ for latent variables (the gray nodes with blue rim.). (b) G-CaRL (Morioka & Hyvärinen, 2023) assumes that observed variables can be grouped according to which latent variables they are connected to (the blue dotted lines), (c) IDOL requires a **sparse** latent causal process (the blue solid lines).

et al., 2019). To seek identifiability in an unsupervised manner, other researchers (Ng et al., 2024; Zheng & Zhang, 2024) propose the structural sparsity on the generation process from latent sources to observation. However, these methods usually do not have instantaneous relations, e.g., the latent variables are mutually independent conditional on their time-delayed parents at previous timestamps. Hence, further assumptions are used for identifiability. For example, Lippe et al. (2023) demonstrate identifiability by assuming that there exist interventions on latent variables, and Morioka & Hyvärinen (2023) yield identifiability with the assumption of the grouping of observational variables. Recently, Zhang et al. (2024) use the sparse structures of latent variables to achieve identifiability of static data under distribution shift. Beyond it, in this work, we demonstrate that temporal identifiability benefits from the sparsity constraint on the causal influences in the latent causal processes by employing the temporal contextual information, i.e., the variables across multiple timestamps. Please refer to Appendix A and D for further discussion of related works, including the identification of latent variables and instantaneous time series.

Although these methods (Lippe et al., 2023; Morioka & Hyvärinen, 2023) have demonstrated the identifiability of latent variables with instantaneous dependencies, they often impose strict assumptions on the latent variables and observations. Specifically, as shown in Figure 1 (a), iCITRIS (Lippe et al., 2023) reaches identifiability by assuming that interventions $I_t$, i.e., the gray nodes with blue rim, on the latent variables, which is expensive and difficult in practice. For instance, considering the joints as the latent variables of a human motion process, it is almost impossible to intervene in the joints of the knee when a human is walking. Moreover, as shown in Figure 1 (b), G-CaRL (Morioka & Hyvärinen, 2023) employs the grouping of observations to identify the latent variables, i.e., the blue dotted lines. However, the grouping of observations is usually expensive, for example, grouping the sensors on a human body from the joints of skeletons requires massive of expert knowledge and labor. Therefore, it is urgent to find a more general approach with an appropriate assumption to identify the latent causal process with instantaneous dependencies.

To identify the latent variables in more relaxed assumptions, we present a general **ID**entification framework for instantane**O**us **L**atent dynamics (**IDOL**) with the only assumption of the sparse causal influences in the latent processed named "sparse latent process" as shown in Figure 1 (c), where the transition process denoted by the blue solid lines is sparse. It is worth noting that the sparse latent process assumption is common and naturally appears in many practical time series data. In the example of human motion forecasting, a skeleton of the human body, which can be considered to be a latent causal process, is usually sparse. This is because there are a few connections among the joints of the human body. Based on the *latent process sparsity*, we first establish the relationship between the ground truth and the estimated latent variables by employing sufficient variability. Sequentially, we propose the sparse latent process assumption to achieve the component-wise identifiability of latent variables by leveraging the contextual information of time series data, which implies the

identification of Markov equivalence class. Furthermore, we can extend to the identification of the causal graph when the endpoints of instantaneous edges do not share identical time-delayed parents, i,e, the example in Figure 1 (c) precisely satisfies this condition. Building upon this identifiability framework, we develop the **IDOL** model, which involves a variational inference neural architecture to reconstruct the latent variables and a gradient-based sparsity penalty to encourage the sparse causal influences in the latent process for identifiability. Evaluation results on the simulation datasets support the theoretical claims and experiment results on several benchmark datasets for time series forecasting highlight the effectiveness of our method.

## 2 PROBLEM SETUP

### 2.1 TIME SERIES GENERATIVE MODEL UNDER INSTANTANEOUS LATENT CAUSAL PROCESS

We begin with the data generation process as shown in Figure 1 (c) where the latent causal process is sparse. To facilitate clarity, we adopt the terminology widely used in ICA literature like observation $\mathbf{x}_t$, latent variables $\mathbf{z}_t$ and mixing function $g$ (Hyvarinen & Morioka, 2016). Suppose that we have time series data with discrete timestamps $\mathbf{X} = \{\mathbf{x}_1, \cdots, \mathbf{x}_t, \cdots, \mathbf{x}_T\}$, where $\mathbf{x}_t$ is generated from latent variables $\mathbf{z}_t \in \mathcal{Z} \subseteq \mathbb{R}^n$ via an invertible and nonlinear mixing function $\mathbf{g}$. We have:

$$\mathbf{x}_t = \mathbf{g}(\mathbf{z}_t). \tag{1}$$

Specifically, the $i$-th dimension of the latent variable $z_{t,i}$ is generated via the latent causal process, which is assumed to be related to the time-delayed parent variables $\mathrm{Pa}_d(z_{t,i})$ and the instantaneous parents $\mathrm{Pa}_e(z_{t,i})$. Formally, it can be written via a structural equation model (Neuberg, 2003) as

$$z_{t,i} = f_i(\mathrm{Pa}_d(z_{t,i}), \mathrm{Pa}_e(z_{t,i}), \epsilon_{t,i}) \quad \text{with} \quad \epsilon_{t,i} \sim p_{\epsilon_i}, \tag{2}$$

where $\epsilon_{t,i}$ denotes the temporally and spatially independent noise extracted from a distribution $p_{\epsilon_i}$. To better understand this data generation process, we take an example of human motion forecasting, where the joint positions can be considered as latent variables. In this case, the rigid body effects among joints result in instantaneous effects, and the human movement trajectory recorded by the sensor are observed variables. As a result, we can consider the process from latent joint positions to movement trajectory as the mixing process.

#### 2.1.1 IDENTIFIABILITY OF LATENT CAUSAL PROCESSES

Based on the aforementioned generation process, we further provide the definition of the identifiability of latent causal process with instantaneous dependency in Definition 1. Moreover, if the estimated latent processes can be identified at least up to permutation and component-wise invertible transformation, the latent causal relations are also immediately identifiable up to a Markov equivalence class (Spirtes et al., 2001). We further show how to go beyond the Markov equivalence class and identify the instantaneous causal relations with a mild assumption in Theorem 3.

**Definition 1** (Identifiable Latent Causal Process (Yao et al., 2022; 2021))**.** *Let* $\mathbf{X} = \{\mathbf{x}_1, \ldots, \mathbf{x}_T\}$ *be a sequence of observed variables generated by the true latent causal processes specified by* $(f_i, p(\epsilon_i), \mathbf{g})$ *given in Equation (1) and (2). A learned generative model* $(\hat{f}_i, \hat{p}(\epsilon_i), \hat{\mathbf{g}})$ *is observational equivalent to* $(f_i, p(\epsilon_i), \mathbf{g})$ *if the model distribution* $p_{\hat{f}_i, \hat{p}_\epsilon, \hat{\mathbf{g}}}(\{\mathbf{x}\}_{t=1}^T)$ *matches the data distribution* $p_{f_i, p_\epsilon, \mathbf{g}}(\{\mathbf{x}\}_{t=1}^T)$ *for any value of* $\{\mathbf{x}\}_{t=1}^T$. *We say latent causal processes are identifiable if observational equivalence can lead to a version of the generative process up to a permutation* $\pi$ *and component-wise invertible transformation* $\mathcal{T}$:

$$p_{\hat{f}_i, \hat{p}_{\epsilon_i}, \hat{\mathbf{g}}}(\{\mathbf{x}\}_{t=1}^T) = p_{f_i, p_{\epsilon_i}, \mathbf{g}}(\{\mathbf{x}\}_{t=1}^T) \;\; \Rightarrow \;\; \hat{\mathbf{g}} = \mathbf{g} \circ \pi \circ \mathcal{T}. \tag{3}$$

Once the mixing process gets identified, the latent variables will be immediately identified up to permutation and component-wise invertible transformation:

$$\hat{\mathbf{z}}_t = \hat{\mathbf{g}}^{-1}(\mathbf{x}_t) = \left(\mathcal{T}^{-1} \circ \pi^{-1} \circ \mathbf{g}^{-1}\right)(\mathbf{x}_t) = \left(\mathcal{T}^{-1} \circ \pi^{-1}\right)\left(\mathbf{g}^{-1}(\mathbf{x}_t)\right) = \left(\mathcal{T}^{-1} \circ \pi^{-1}\right)(\mathbf{z}_t). \tag{4}$$

## 3 IDENTIFICATION RESULTS FOR LATENT CAUSAL PROCESS

Given the definition of identification of latent causal process, we show how to identify the latent causal process in Figure 1 (c) under a sparse latent process. Please note that our theorem is discussed

under Markov Network. For more details, please refer to Appendix B.6. Specifically, we first leverage the connection between conditional independence and cross derivatives (Lin, 1997) and the sufficient variability of temporal data to discover the relationships between the estimated and the true latent variables, which is shown in Theorem 1. Moreover, we establish the identifiability result of latent variables by enforcing the sparse causal influences, as shown in Theorem 2. Moreover, we also show that the existing identifiability results of the temporally causal representation learning methods are a special case of our **IDOL** method, which is shown in Corollary 1. Finally, under a mild assumption, we achieve identification of the causal graph underlying the latent causal process, as shown in Theorem 3.

## 3.1 RELATIONSHIPS BETWEEN GROUND-TRUTH AND ESTIMATED LATENT VARIABLES

In this section, we figure out the relationships between ground truth and estimated latent variables under a temporal scenario with instantaneous dependence. In order to incorporate contextual information to aid in the identification of latent variables $\mathbf{z}_t$, latent variables of $L$ consecutive timestamps including $\mathbf{z}_t$, are taken into consideration. Without loss of generality, we consider a simplified case, where the length of the sequence is 2, i.e., $L = 2$, and the time lag is 1, i.e., $\tau = 1$. Please refer to Appendix B.2 for the general case of multiple lags and sequence lengths.

**Theorem 1.** *For a series of observations $\mathbf{x}_t \in \mathbb{R}^n$ and estimated latent variables $\hat{\mathbf{z}}_t \in \mathbb{R}^n$ with the corresponding process $\hat{f}_i, \hat{p}(\epsilon), \hat{g}$, where $\hat{g}$ is invertible, suppose that the process subject to observational equivalence $\mathbf{x}_t = \hat{\mathbf{g}}(\hat{\mathbf{z}}_t)$. Let $\mathbf{c}_t \triangleq \{\mathbf{z}_{t-1}, \mathbf{z}_t\} \in \mathbb{R}^{2n}$ and that $\mathcal{M}_{\mathbf{c}_t}$ be the variable set of two consecutive timestamps and the corresponding Markov network respectively. Suppose that the following assumptions hold:*

- *A1 (Smooth and Positive Density): The conditional probability function of the latent variables $\mathbf{c}_t$ is smooth and positive, i.e., $p(\mathbf{c}_t | \mathbf{z}_{t-2})$ is third-order differentiable and $p(\mathbf{c}_t | \mathbf{z}_{t-2}) > 0$ over $\mathbb{R}^{2n}$,*

- *A2 (Sufficient Variability): Denote $|\mathcal{M}_{\mathbf{c}_t}|$ as the number of edges in Markov network $\mathcal{M}_{\mathbf{c}_t}$. Let*

$$
w(m) = \Big( \frac{\partial^3 \log p(\mathbf{c}_t | \mathbf{z}_{t-2})}{\partial c_{t,1}^2 \partial z_{t-2,m}}, \cdots, \frac{\partial^3 \log p(\mathbf{c}_t | \mathbf{z}_{t-2})}{\partial c_{t,2n}^2 \partial z_{t-2,m}} \Big) \oplus
$$
$$
\Big( \frac{\partial^2 \log p(\mathbf{c}_t | \mathbf{z}_{t-2})}{\partial c_{t,1} \partial z_{t-2,m}}, \cdots, \frac{\partial^2 \log p(\mathbf{c}_t | \mathbf{z}_{t-2})}{\partial c_{t,2n} \partial z_{t-2,m}} \Big) \oplus \Big( \frac{\partial^3 \log p(\mathbf{c}_t | \mathbf{z}_{t-2})}{\partial c_{t,i} \partial c_{t,j} \partial z_{t-2,m}} \Big)_{(i,j) \in \mathcal{E}(\mathcal{M}_{\mathbf{c}_t})}, \quad (5)
$$

*where $\oplus$ denotes concatenation operation and $(i,j) \in \mathcal{E}(\mathcal{M}_{\mathbf{c}_t})$ denotes all pairwise indice such that $c_{t,i}, c_{t,j}$ are adjacent in $\mathcal{M}_{\mathbf{c}_t}$. For $m \in [1, \cdots, n]$, there exist $4n + |\mathcal{M}_{\mathbf{c}_t}|$ different values of $\mathbf{z}_{t-2,m}$, such that the $4n + |\mathcal{M}_{\mathbf{c}_t}|$ values of vector functions $w(m)$ are linearly independent.*

*Then for any two different entries $\hat{c}_{t,k}, \hat{c}_{t,l}$ of $\hat{\mathbf{c}}_t \in \mathbb{R}^{2n}$ that are **not adjacent** in the Markov network $\mathcal{M}_{\hat{\mathbf{c}}_t}$ over estimated $\hat{\mathbf{c}}_t$,*
*(i) Each ground-truth latent variable $c_{t,i}$ of $\mathbf{c}_t \in \mathbb{R}^{2n}$ is a function of at most one of $\hat{c}_k$ and $\hat{c}_l$,*
*(ii) For each pair of ground-truth latent variables $c_{t,i}$ and $c_{t,j}$ of $\mathbf{c}_t \in \mathbb{R}^{2n}$ that are **adjacent** in $\mathcal{M}_{\mathbf{c}_t}$ over $\mathbf{c}_t$, they can not be a function of $\hat{c}_{t,k}$ and $\hat{c}_{t,l}$ respectively.*

**Proof Sketch.** The proof can be found in Appendix B.1. First, we establish a bijective transformation between the ground-truth $\mathbf{z}_t$ and the estimated $\hat{\mathbf{z}}_t$ to connect them together. Next, we utilize the structural properties of the ground-truth Markov network $\mathcal{M}_{\mathbf{c}_t}$ to constrain the structure of the estimated Markov network $\mathcal{M}_{\hat{\mathbf{c}}_t}$ through the connection between conditional independence and cross derivatives (Lin, 1997), i.e., $c_{t,i} \perp c_{t,j} | \mathbf{c}_t \backslash \{c_{t,i}, c_{t,j}\}$ implies $\frac{\partial^2 \log p(\mathbf{c}_t)}{\partial c_{t,i} \partial c_{t,j}} = 0$. By introducing the sufficient variability assumption, we further construct a linear system with a full rank coefficient matrix to ensure that the only solution holds, i.e.,

$$
\frac{\partial c_{t,i}}{\partial \hat{c}_{t,k}} \cdot \frac{\partial c_{t,i}}{\partial \hat{c}_{t,l}} = 0, \quad \frac{\partial c_{t,i}}{\partial \hat{c}_{t,k}} \cdot \frac{\partial c_{t,j}}{\partial \hat{c}_{t,l}} = 0, \quad \frac{\partial^2 c_{t,i}}{\partial \hat{c}_{t,k} \partial \hat{c}_{t,l}} = 0, \quad (6)
$$

where $\frac{\partial c_{t,i}}{\partial \hat{c}_{t,k}} \cdot \frac{\partial c_{t,i}}{\partial \hat{c}_{t,l}} = 0$ and $\frac{\partial c_{t,i}}{\partial \hat{c}_{t,k}} \cdot \frac{\partial c_{t,j}}{\partial \hat{c}_{t,l}} = 0$ correspond to statement (i) statement (ii), respectively.

Theorem 1 provides an insight, such that when observational equivalence holds, the variables that are directly related in the true Markov network must be functions of directly related variables in

the estimated Markov network. Please note that $\tau$ and $L$ can be easily generalized to any value by making some modifications on the assumption $A2$. The $\tau$ timestamps that are conditioned on provide sufficient changes, and $L$ sequence length provides a sparse structure, which will be discussed in Theorem 2. A detailed discussion is given in Appendix B.2.

## 3.2 IDENTIFICATION OF LATENT VARIABLES

In this subsection, we demonstrate that given further sparsity assumptions, the latent Markov network over $\mathbf{c}_t$ and the latent variables are also identifiable. For a better explanation of the identifiability of latent variables, we first introduce the definition of the **Intimate Neighbor Set** (Zhang et al., 2024).

**Definition 2** (Intimate Neighbor Set (Zhang et al., 2024)). *Consider a Markov network $\mathcal{M}_Z$ over variables set $Z$, and the intimate neighbor set of variable $z_{t,i}$ is*

$$\Psi_{\mathcal{M}_{\mathbf{c}_t}}(c_{t,i}) \triangleq \{c_{t,j} | c_{t,j} \text{ is adjacent to } c_{t,i}, \text{ and it is also adjacent to all other neighbors of } c_{t,i}, c_{t,j} \in \mathbf{c}_t \backslash \{c_{t,i}\}\}.$$

In other words, $\Psi$ contains all neighbors, iff all neighbors of $c_{t,i}$ are also in the same **unique** clique as $c_{t,i}$, else $\Psi = \emptyset$. We further provide an example in the Appendix B.8.

Based on the conclusions of Theorem 1, we consider 3 consecutive timestamps, i.e., $\mathbf{c}_t = \{\mathbf{z}_{t-1}, \mathbf{z}_t, \mathbf{z}_{t+1}\}$, where the identifiability of $\mathbf{z}_t$ can be assured with the help of the contextual information $\mathbf{z}_{t-1}$ and $\mathbf{z}_{t+1}$.

**Theorem 2.** *(Component-wise Identification of Latent Variables with instantaneous dependencies.) Suppose that the observations are generated instantaneous latent process, and $\mathcal{M}_{\mathbf{c}_t}$ is the Markov network over $\mathbf{c}_t = \{\mathbf{z}_{t-1}, \mathbf{z}_t, \mathbf{z}_{t+1}\} \in \mathbb{R}^{3n}$. Except for the smooth, positive density and sufficient variability assumptions, we further make the following assumption:*

- *A3 (Sparse Latent Process ): For any $z_{it} \in \mathbf{z}_t$, the intimate neighbor set of $z_{it}$ is an empty set.*

*When the observational equivalence is achieved with the minimal number of edges of the estimated Markov network of $\mathcal{M}_{\hat{\mathbf{c}}_t}$, then we have the following two statements:*

*(i) The estimated Markov network $\mathcal{M}_{\hat{\mathbf{c}}_t}$ is isomorphic [1] to the ground-truth Markov network $\mathcal{M}_{\mathbf{c}_t}$.*

*(ii) There exists a permutation $\pi$ of the estimated latent variables, such that $z_{it}$ and $\hat{z}_{\pi(i)t}$ is one-to-one corresponding, i.e., $z_{it}$ is component-wise identifiable.*

**Proof Sketch.** Given the fact that there always exists a row permutation for each invertible matrix such that the permuted diagonal entries are non-zero (Zheng & Zhang, 2024), we utilize the conclusion of Theorem 1 to demonstrate that any edge present in the true Markov network will necessarily exist in the estimated one. Furthermore, when the number of estimated edges reaches a minimum, the identifiability of Markov network can be achieved. Finally, we once again leverage Theorem 1 to illustrate that the permutation which allows the Markov network to be identified can further lead to a component-wise level identifiability of latent variables. The detailed proof and discussion of assumptions are given in Appendix B.3 and C, respectively.

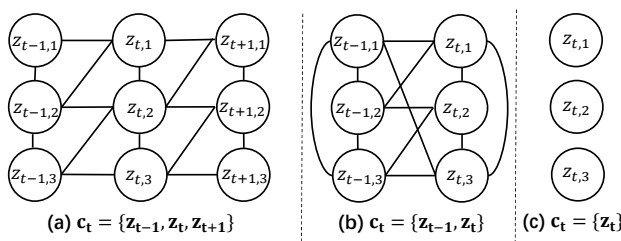

(a) $\mathbf{c_t} = \{\mathbf{z_{t-1}}, \mathbf{z_t}, \mathbf{z_{t+1}}\}$    (b) $\mathbf{c_t} = \{\mathbf{z_{t-1}}, \mathbf{z_t}\}$    (c) $\mathbf{c_t} = \{\mathbf{z_t}\}$

Figure 2: Examples of Markov networks with different types of contextual information that satisfy the sparse latent process assumption. Case (i) uses both historical and future information for identifiability. Case (ii) uses only historical information. Case (iii) that does not include instantaneous dependencies can identify latent variables without any contextual information, which degenerates to TDRL (Yao et al., 2022). Please note that in all cases, we show only the Markov net of $\mathbf{c}_t$ conditioned on previous timestamps while omitting the conditions for simplicity.

**Discussion.** Intuitively, Theorem 2 tells us that any latent variable $z_{t,i}$ is identifiable when its intimate set is empty, which benefits from the sparse causal influence. Compared with the existing

---
[1]Please refer to the definition of isomorphism of Markov networks in Appendix B.7.

Table 1: The summary of related work of causal representation learning.

| | No Intervention | No Grouping | Stationarity | Instantaneous Effect | Temporal Data |
|---|---|---|---|---|---|
| IDOL | ✓ | ✓ | ✓ | ✓ | ✓ |
| Yao et al. (2022) | ✓ | ✓ | ✓ | ✗ | ✓ |
| Morioka & Hyvärinen (2023) | ✓ | ✗ | ✓ | ✓ | ✓ |
| Lippe et al. (2023) | ✗ | ✓ | ✓ | ✓ | ✓ |
| Zhang et al. (2024) | ✓ | ✓ | ✗ | ✓ | ✗ |

identifiability results for instantaneous dependencies like intervention or grouping observations, this assumption is more practical and reasonable in real-world time series datasets.

One more thing to note is that it is not necessary to utilize all contextual information for identifiability. In fact, as long as there exist consecutive timestamps where the intimate sets of all variables in $\mathbf{z}_t$ are empty, the identifiability can be established.

**Corollary 1.** *(**General Case for Component-wise Identification.**) Suppose that the observations are generated by Equation (1) and (2), and there exists $\mathbf{c}_t = \{\mathbf{z}_{t-a}, \cdots, \mathbf{z}_t, \cdots, \mathbf{z}_{t+b}\} \in \mathbb{R}^{(a+b+1) \times n}$ with the corresponding Markov network $\mathcal{M}_{\mathbf{c}_t}$. Suppose assumptions A1 and A2 hold true, and for any $z_{t,i} \in \mathbf{z}_t$, the intimate neighbor set of $z_{t,i}$ is an empty set. When the observational equivalence is achieved with the minimal number of edges of the estimated Markov network of $\mathcal{M}_{\hat{\mathbf{c}}_t}$, there must exist a permutation $\pi$ of the estimated latent variables, such that $z_{t,i}$ and $\hat{z}_{t,\pi(i)}$ is one-to-one corresponding, i.e., $z_{t,i}$ is component-wise identifiable.*

**Discussion.** In this part, we give a further discussion on how contextual information can help to identify current latent variables. Let us take Figure 2 (i) as an example, when we consider the current latent variables $z_{t,1}, z_{t,2}$ and $z_{t,3}$ only, it is easy to see that neither $z_{t,1}$ nor $z_{t,3}$ are identifiable since both of their intimate set are $\{z_{t,2}\}$. However, when the historical information $\mathbf{z}_{t-1}$ is taken into consideration, where $z_{t-1,1}$ affects only $z_{t,1}$ but not $z_{t,2}$, then $z_{t,1}$ will immediately be identified since the intimate set of $z_{t,1}$ will be empty thanks to the unique influence from $z_{t-1,1}$. Similarly, with a future variable $z_{t+1,3}$ which is influenced by $z_{t,3}$ but not $z_{t,1}$, then $z_{t,3}$ can be identified as well. Besides, under some circumstances, only part of the contextual information is needed. Therefore, the length of the required time window $L$ can be reduced as well, which leads to a more relaxed requirement of assumption A2, according to Theorem 1. As shown in Figure 2 (ii), with only historical information, the identifiability of $\mathbf{z}_t$ can already be achieved. Furthermore, some existing identifiability results (Yao et al., 2021; 2022) can be considered as a special case of our theoretical results when the instantaneous effects vanish. For example, Figure 2 (iii) provides the example of TDRL (Yao et al., 2022), where all intimate sets are empty naturally, and the identifiability is assured.

Finally, we explore under what circumstances the causal graph of latent variables is also identifiable.

**Theorem 3.** *(**Identification of Latent Causal Process.**) Suppose that the observations are generated by Equation (1)-(2), and that $\mathcal{M}_{\mathbf{c}_t}$ is the Markov network over $\mathbf{c}_t = \{\mathbf{z}_{t-1}, \mathbf{z}_t\} \in \mathbb{R}^{2n}$. Suppose that all assumptions for Theorem 2 hold. We further make the following assumption: for any pair of adjacent latent variables $z_{t,i}, z_{t,j}$ at time step $t$, their time-delayed parents are not identical, i.e., $Pa_d(z_{t,i}) \neq Pa_d(z_{t,j})$. Then, the causal graph of the latent causal process is identifiable.*

**Proof Sketch.** We first establish the identification of the skeleton of the causal graph using the results of Theorem 2. The directions of time-delayed edges are fully determined since time can only move forward. Finally, we leverage the v-structure to identify the instantaneous edges. A detailed proof is provided in Appendix B.5.

## 3.3 COMPARASION WITH EXISTING METHODS

The application of real-world scenarios presents numerous new opportunities and challenges. Compared to existing methods of temporally causal representation learning with instantaneous effects, our approach significantly relaxes the constraints on theoretical assumptions, as shown in Table 1. TDRL (Yao et al., 2022) requires that there are no instantaneous effects during the transition process, which can be considered as a special case of our method. G-CaRL (Morioka & Hyvärinen, 2023) and iCITRIS (Lippe et al., 2023) require extra intervention or grouping information, which is not necessary in our method. Zhang et al. (2024) focus on non-temporal scenarios and require extra domain labels. Instead, IDOL can handle the stationary temporal case. Besides, when the instantaneous effect

is dense, which is beyond the capacity of Zhang et al. (2024), IDOL can still utilize the contextual information for better identifiability as long as the temporal transition is sparse.

# 4 IDENTIFIABLE INSTANTANEOUS LATENT DYNAMIC MODEL

## 4.1 TEMPORALLY VARIATIONAL INFERENCE ARCHITECTURE

Based on the data generation process, we derive the evidence lower bound (ELBO) as shown in Equation (7) for general time series modeling. Please refer to Appendix E.2 for the derivation details of the EBLO for time series forecasting.

$$ELBO = \underbrace{\mathbb{E}_{q(\mathbf{z}_{1:T}|\mathbf{x}_{1:T})} \ln p(\mathbf{x}_{1:T}|\mathbf{z}_{1:T})}_{\mathcal{L}_r}$$
$$- \underbrace{D_{KL}(q(\mathbf{z}_{1:T}|\mathbf{x}_{1:T})||p(\mathbf{z}_{1:T}))}_{\mathcal{L}_K}, \quad (7)$$

where $D_{KL}$ denotes the KL divergence. We use posterior $q(\mathbf{z}_{1:T}|\mathbf{x}_{1:T})$ to approximate the prior $p(\mathbf{z}_{1:T})$. Besides, $p(\mathbf{x}_{1:T}|\mathbf{z}_{1:T})$ is used to reconstruct the observations. The encoder and the decoder can be formalized as follows:

$$\mathbf{z}_{1:T} = \phi(\mathbf{x}_{1:T}), \quad \hat{\mathbf{x}} = \psi(\mathbf{z}_{1:T}), \quad (8)$$

where $\phi$ and $\psi$ denote the neural architecture based on convolution neural networks. Please refer to Table A4 for the implementation details of $\phi$ and $\psi$.

Figure 3: The framework of the **IDOL** model. The encoder and decoder are used for the extraction of latent variables and observation reconstruction. The prior network is used for prior distribution estimation, and $L_s$ denotes the gradient-based sparsity penalty. The solid and dashed arrows denote the forward and backward propagation.

Based on theoretical results, we develop the **IDOL** model as shown in Figure 3, which is built on the variational inference to model the distribution of observations. To estimate the prior distribution and enforce the independent noise assumption, we devise the prior networks. Moreover, we employ a gradient-based sparsity penalty to promote the sparse causal influence.

## 4.2 PRIOR ESTIMATION

To enforce the independence of noise in Equation (2), we minimize the Kullback-Leibler (KL) divergence between the posterior distribution $\prod_i p(z_{t,i}|\text{Pa}_d(z_{t,i}), \text{Pa}_e(z_{t,i}), \mathbf{x}_t)$ and a prior $\prod_i p(z_{t,i}|\text{Pa}_d(z_{t,i}), \text{Pa}_e(z_{t,i}))$. This minimization implies that latent variables are mutually independent and conditioned on their historical and instantaneous parents. However, since the prior distribution may have any arbitrary density function, it is difficult to estimate such a prior. To solve this challenge, we follow Chen et al. (2024a), Yao et al. (2022) and propose the prior networks. Specifically, we first let $r_i$ be an inversed function of $f$ from Equation (2) that are implemented by normalizing flow. Papamakarios et al. (2021) take the estimated latent variables as input to estimate the noise term $\hat{\epsilon}_i$, i.e. $\hat{\epsilon}_{t,i} = r_i(\hat{\mathbf{z}}_t, \hat{\mathbf{z}}_{t-1})^2$. And each $r_i$ is implemented by Multi-layer Perceptron networks (MLPs). Sequentially, we devise a transformation $\kappa := \{\hat{\mathbf{z}}_{t-1}, \hat{\mathbf{z}}_t\} \to \{\hat{\mathbf{z}}_{t-1}, \hat{\epsilon}_t\}$, whose Jacobian can be formalized as $\mathbf{J}_\kappa = \begin{pmatrix} \mathbb{I} & 0 \\ \mathbf{J}_d & \mathbf{J}_e \end{pmatrix}$, where $\mathbf{J}_d = \left(\frac{\partial r_i}{\partial \hat{z}_{t-1,i}}\right)$ and $\mathbf{J}_e = \left(\frac{\partial r_i}{\partial \hat{z}_{t,i}}\right)$. Hence we have Equation (9) via the change of variables formula.

$$\log p(\hat{\mathbf{z}}_t, \hat{\mathbf{z}}_{t-1}) = \log p(\hat{\mathbf{z}}_{t-1}, \epsilon_t) + \log |\frac{\partial r_i}{\partial z_{t,i}}|. \quad (9)$$

According to the generation process, the noise $\epsilon_{t,i}$ is independent with $\mathbf{z}_{t-1}$, so we can enforce the independence of the estimated noise term $\hat{\epsilon}_{t,i}$. And Equation (9) can be further rewritten as

$$\log p(\hat{\mathbf{z}}_{1:T}) = \log p(\hat{z}_1) + \sum_{\tau=2}^{T} \left( \sum_{i=1}^{n} \log p(\hat{\epsilon}_{\tau,i}) + \sum_{i=1}^{n} \log |\frac{\partial r_i}{\partial z_{t,i}}| \right), \quad (10)$$

where $p(\hat{\epsilon}_{\tau,i})$ is assumed to follow a Gaussian distribution. Please refer to Appendix E.1 for more details of the prior derivation.

---

[2]We use the superscript symbol to denote estimated latent variables

### 4.3 GRADIENT-BASED SPARSITY REGULARIZATION

Ideally, the MLPs-based architecture $r_i$ can capture the causal structure of latent variables by restricting the independence of noise $\hat{\epsilon}_{t,i}$. However, without any further constraint, $r_i$ may bring redundant causal edges from $\hat{\mathbf{z}}_{t-1}, \hat{\mathbf{z}}_{t,[n]\backslash i}$ to $\hat{z}_{t,i}$, leading to the incorrect estimation of prior distribution and further the suboptimization of ELBO. As mentioned in Subsection 4.2, $\mathbf{J}_d$ and $\mathbf{J}_e$ intuitively denote the time-delayed and instantaneous causal structures of latent variables, since they describe how the $\hat{\mathbf{z}}_{t-1}, \hat{\mathbf{z}}_{t,[n]\backslash i}$ contribute to $\hat{z}_{t,i}$, which motivate us to remove these redundant causal edges with a sparsity regularization term $\mathcal{L}_S$ by simply applying the $\mathcal{L}_1$ on $\mathbf{J}_d$ and $\mathbf{J}_e$. Formally, we have

$$\mathcal{L}_S = ||\mathbf{J}_d||_1 + ||\mathbf{J}_e||_1., \tag{11}$$

where $|| * ||_1$ denotes the $\mathcal{L}_1$ Norm of a matrix. By employing the gradient-based sparsity penalty on the estimated latent causal processes, we can indirectly restrict the sparsity of Markov networks to satisfy the sparse latent process. Finally, the total loss of the **IDOL** model can be formalized as:

$$\mathcal{L}_{total} = -\mathcal{L}_r - \alpha\mathcal{L}_K + \beta\mathcal{L}_S, \tag{12}$$

where $\alpha, \beta$ denote the hyper-parameters.

## 5 EXPERIMENTS

### 5.1 EXPERIMENTS ON SIMULATION DATA

#### 5.1.1 EXPERIMENTAL SETUP

**Data Generation.** We generate the simulated time series data with the fixed latent causal process as introduced in Equations (1)-(2) and Figure 1 (c). To better evaluate the proposed theoretical results, we provide six synthetic datasets from A to F with 3, 5, 8, 8, 8, 16 latent variables, respectively. Dataset D contains no instantaneous effects, which degenerate to the TDRL (Yao et al., 2022) setting as introduced in Figure 2 (iii). Dataset E has a dense latent causal process that violates the assumption A3(Latent Process Sparsity). All datasets except E satisfy the assumption A3. Please refer to Appendix F.1.1 for the details of data generation and evaluation metrics.

**Baselines.** To evaluate the effectiveness of our method, we consider the following compared methods. First, we consider the standard $\beta$-VAE (Higgins et al., 2017) and FactorVAE (Kim & Mnih, 2018), which ignores historical information and auxiliary variables. Moreover, we consider TDRL (Yao et al., 2022), iVAE (Khemakhem et al., 2020a), TCL (Hyvarinen & Morioka, 2016), PCL (Hyvarinen & Morioka, 2017), and SlowVAE (Klindt et al., 2020), which use temporal information but do not assume instantaneous dependency on latent processes. Finally, we consider the iCITRIS (Lippe et al., 2023) and G-CaRL (Morioka & Hyvärinen, 2023), which are devised for latent causal processes with instantaneous dependencies but require interventions or grouping of observations. As for G-CaRL, we follow the implementation description in the original paper and assign random grouping for observations. As for iCITRIS, we assign a random intervention variable for latent variables. We repeat each method over three random seeds and report the average results.

#### 5.1.2 RESULTS AND DISCUSSION

**Quantitative Results:** Experiment results of the simulation datasets are shown in Table 2. The proposed **IDOL** model achieves the highest MCC performance, reflecting that our method can identify the latent variables under the temporally latent process with instantaneous dependency. According to the experiment results, we can obtain the following conclusions: 1) our **IDOL** method also outperforms G-CaRL and iCITRIS, which leverage grouping and interventions for identifiability of latent variables with instantaneous dependencies, reflecting that our method does not require a stricter assumption for identifiability. 2) Compared with the existing methods for temporal data like TDRL, PCL, and TCL, the experiment results show that our method can identify the general stationary latent process with instantaneous dependencies. 3) According to the results of the dense dataset E, the IDOL cannot well identify the latent variables, validating the boundary of our theoretical results. 4) We further analyze the effectiveness of the sparsity regularization. Please refer to Appendix F.2.4,F.1.4 for more details.

**Qualitative Results:** Figure 4 provides visualization results on dataset A of the latent causal process of IDOL, TDRL, G-CaRL, and iCITRIS. Note that the visualization results of our method are

Table 2: Experiments results of MCC on simulation data. The iCITRIS algorithm encountered an out-of-memory (OOM) issue when running on dataset F due to the high dimensionality of the latent variables, which caused the code to fail to execute. This is indicated by the symbol '-'.

| Datasets | IDOL | TDRL | G-CaRL | iCITRIS | $\beta-$VAE | SlowVAE | iVAE | FactorVAE | PCL | TCL |
|---|---|---|---|---|---|---|---|---|---|---|
| A | **0.9645** | 0.9416 | 0.9059 | 0.8219 | 0.8485 | 0.8512 | 0.6283 | 0.8512 | 0.8659 | 0.8625 |
| B | **0.9142** | 0.8727 | 0.6248 | 0.4120 | 0.4113 | 0.2875 | 0.5545 | 0.2158 | 0.5288 | 0.3311 |
| C | **0.9801** | 0.9001 | 0.5850 | 0.4234 | 0.4093 | 0.3420 | 0.6736 | 0.2417 | 0.3981 | 0.2796 |
| D | 0.9766 | **0.9796** | 0.5455 | 0.3343 | 0.2181 | 0.2641 | 0.3469 | 0.2527 | 0.4806 | 0.2461 |
| E | **0.7869** | 0.7228 | 0.5835 | 0.4646 | 0.4260 | 0.3986 | 0.6071 | 0.2319 | 0.4659 | 0.2881 |
| F | **0.9747** | 0.5899 | 0.5225 | - | 0.4321 | 0.5157 | 0.3176 | 0.2648 | 0.5908 | 0.6678 |

Figure 4: Visualization results of directed acyclic graphs of latent variables of different methods. The first and second rows denote time-delayed and instantaneous causal relationships of latent variables.

generated from $\mathbf{J}_d$ and $\mathbf{J}_e$. According to the experiment results, we can find that the proposed method can reconstruct the latent causal relationship well, which validates the theoretical results. Please note that here, not only the Markov equivalence class but also the causal graph can be identified for dataset A, as shown in Figure 4. Please refer to Appendix B.5 for more details. Moreover, since TDRL employs the conditional independent assumption, it does not reconstruct any instantaneous causal relationships. Besides, without any accurate grouping or intervention information, the G-CaRL and iCITRIS can not reconstruct the correct latent causal relationships. Please refer to Appendix E.3.1 for more details.

## 5.2 EXPERIMENTS ON REAL-WORLD DATA

### 5.2.1 EXPERIMENT SETUP

**Datasets.** To evaluate the effectiveness of our **IDOL** method in real-world scenarios, we conduct experiments on two human motion datasets: Human3.6M (Ionescu et al., 2014) and HumanEva-I (Sigal et al., 2010), which record the location and orientation of local coordinate systems at each joint. We consider these datasets since the joints can be considered as latent variables and they naturally contain instantaneous dependencies. We choose several motions to conduct time series forecasting. For the Human3.6M datasets, we consider 4 motions: Gestures (Ge), Jog (J), CatchThrow (CT), and Walking (W). For HumanEva-I dataset, we consider 6 motions: Discussion (D), Greeting (Gr), Purchases (P), SittingDown (SD), Walking (W), and WalkTogether (WT). For each dataset, we select several motions and partition them into training, validation, and test sets. Please refer the Appendix F.2.1 for the dataset descriptions.

**Baselines.** To evaluate the effectiveness of the proposed IDOL, we consider the following state-of-the-art deep forecasting models for time series forecasting. First, we consider the conventional methods for time series forecasting including Autoformer (Wu et al., 2021), TimesNet (Wu et al., 2022) and MICN (Wang et al., 2022). Moreover, we consider several latest methods for time series analysis like CARD (Wang et al., 2023), FITS (Xu et al., 2024b), and iTransformer (Liu et al., 2024). Finally, we consider the TDRL (Yao et al., 2022). We repeat each experiment over 3 random seeds and publish the average performance. Please refer to Appendix F.2 for more experiment details.

Table 3: MSE and MAE results on the different motions.

| dataset | | Predict Length | IDOL | | TDRL | | CARD | | FITS | | MICN | | iTransformer | | TimesNet | | Autoformer | |
|---|---|---|---|---|---|---|---|---|---|---|---|---|---|---|---|---|---|---|
| | | | MSE | MAE | MSE | MAE | MSE | MAE | MSE | MAE | MSE | MAE | MSE | MAE | MSE | MAE | MSE | MAE |
| Humaneva | G | 100 | **0.0658** | **0.1623** | 0.0729 | 0.1806 | 0.0898 | 0.1999 | 0.0728 | 0.1758 | 0.0781 | 0.1896 | 0.0905 | 0.2013 | 0.0832 | 0.1949 | 0.1039 | 0.2232 |
| | | 125 | **0.0809** | **0.1916** | 0.0878 | 0.1993 | 0.0896 | 0.1985 | 0.0834 | 0.1994 | 0.0832 | 0.1990 | 0.0901 | 0.2000 | 0.0830 | 0.1934 | 0.1010 | 0.2195 |
| | | 150 | **0.0697** | **0.1754** | 0.0724 | 0.1796 | 0.0827 | 0.1894 | 0.0866 | 0.1950 | 0.0715 | 0.1789 | 0.0875 | 0.1965 | 0.0831 | 0.1942 | 0.1006 | 0.2192 |
| | J | 125 | **0.1516** | **0.2479** | 0.2310 | 0.3392 | 0.2836 | 0.3486 | 0.3277 | 0.4058 | 0.1598 | 0.2587 | 0.2306 | 0.3258 | 0.2189 | 0.3071 | 0.3370 | 0.2532 |
| | | 150 | **0.1572** | **0.2632** | 0.2333 | 0.3431 | 0.3614 | 0.3936 | 0.3396 | 0.4142 | 0.1648 | 0.2713 | 0.2874 | 0.3673 | 0.2695 | 0.3526 | 0.3367 | 0.3199 |
| | | 175 | **0.1742** | **0.2786** | 0.2810 | 0.3710 | 0.3938 | 0.4246 | 0.3552 | 0.4329 | 0.1864 | 0.2945 | 0.3074 | 0.3841 | 0.3056 | 0.3707 | 0.3147 | 0.2934 |
| | TC | 25 | **0.0060** | **0.0490** | 0.0086 | 0.0607 | 0.0101 | 0.0614 | 0.0116 | 0.0651 | 0.0086 | 0.0607 | 0.0104 | 0.0619 | 0.0147 | 0.0723 | 0.0254 | 0.1043 |
| | | 50 | **0.0128** | 0.0718 | 0.0151 | 0.0811 | 0.0172 | 0.0801 | 0.0142 | 0.0725 | 0.0158 | 0.0835 | 0.0138 | **0.0711** | 0.0219 | 0.0891 | 0.0263 | 0.1062 |
| | | 75 | 0.0175 | 0.0844 | 0.0191 | 0.0896 | 0.0228 | 0.0686 | 0.0146 | 0.0729 | 0.0185 | 0.0867 | **0.0136** | **0.0701** | 0.0203 | 0.0869 | 0.0279 | 0.1108 |
| | W | 25 | 0.0670 | **0.1338** | 0.0968 | 0.1704 | 0.1010 | 0.1641 | 0.1094 | 0.2117 | 0.0967 | 0.1851 | **0.0604** | 0.1344 | 0.0958 | 0.1710 | 0.0940 | 0.1767 |
| | | 50 | **0.1183** | **0.1814** | 0.1461 | 0.2172 | 0.2387 | 0.2578 | 0.2152 | 0.3089 | 0.1521 | 0.2228 | 0.1245 | 0.2043 | 0.1730 | 0.2389 | 0.3093 | 0.3498 |
| | | 75 | **0.1977** | **0.2543** | 0.2091 | 0.2642 | 0.4777 | 0.3673 | 0.3156 | 0.3817 | 0.2124 | 0.2706 | 0.2239 | 0.2784 | 0.2202 | 0.2884 | 0.3854 | 0.4009 |
| Human | D | 125 | **0.0071** | **0.0485** | 0.0074 | 0.0509 | 0.0080 | 0.0510 | 0.0085 | 0.0523 | 0.0080 | 0.0524 | 0.0076 | 0.0486 | 0.0097 | 0.0568 | 0.0104 | 0.0586 |
| | | 250 | **0.0094** | **0.0563** | 0.0096 | 0.0590 | 0.0117 | 0.0600 | 0.0114 | 0.0590 | 0.0107 | 0.0598 | 0.0112 | 0.0581 | 0.0133 | 0.0643 | 0.0134 | 0.0656 |
| | | 375 | **0.0102** | 0.0638 | 0.0106 | 0.0621 | 0.0138 | 0.0645 | 0.0124 | 0.0615 | 0.0109 | **0.0605** | 0.0126 | 0.0617 | 0.0152 | 0.0675 | 0.0141 | 0.0678 |
| | G | 125 | **0.0120** | **0.0641** | 0.0167 | 0.0757 | 0.0197 | 0.0763 | 0.0239 | 0.0866 | 0.0144 | 0.0703 | 0.0137 | 0.0649 | 0.0195 | 0.0784 | 0.0217 | 0.0845 |
| | | 250 | **0.0158** | **0.0808** | 0.0218 | 0.0880 | 0.0283 | 0.0932 | 0.0298 | 0.0982 | 0.0203 | 0.0847 | 0.0217 | 0.0832 | 0.0277 | 0.0933 | 0.0287 | 0.0974 |
| | | 375 | **0.0226** | **0.0902** | 0.0234 | 0.0914 | 0.0295 | 0.0970 | 0.0304 | 0.1011 | 0.0233 | 0.0912 | 0.0263 | 0.0920 | 0.0311 | 0.0988 | 0.0319 | 0.1029 |
| | P | 125 | **0.0203** | **0.0778** | 0.0233 | 0.0866 | 0.0247 | 0.0837 | 0.0327 | 0.0987 | 0.0237 | 0.0862 | 0.0228 | 0.0793 | 0.0308 | 0.0939 | 0.0400 | 0.1108 |
| | | 250 | **0.0296** | **0.1003** | 0.0303 | 0.1060 | 0.0407 | 0.1109 | 0.0426 | 0.1168 | 0.0358 | 0.1146 | 0.0434 | 0.1182 | 0.0554 | 0.1337 | 0.0546 | 0.1361 |
| | | 375 | **0.0324** | **0.1104** | 0.0333 | 0.1148 | 0.0480 | 0.1268 | 0.0509 | 0.1315 | 0.0364 | 0.1199 | 0.0495 | 0.1312 | 0.0595 | 0.1439 | 0.0638 | 0.1498 |
| | SD | 125 | **0.0142** | **0.0709** | 0.0157 | 0.0777 | 0.0175 | 0.0772 | 0.0209 | 0.0889 | 0.0163 | 0.0802 | 0.0162 | 0.0735 | 0.0236 | 0.0948 | 0.0279 | 0.1086 |
| | | 250 | **0.0250** | **0.1046** | 0.0251 | 0.1050 | 0.0313 | 0.1113 | 0.0331 | 0.1176 | 0.0256 | 0.1069 | 0.0289 | 0.1064 | 0.0355 | 0.1212 | 0.0378 | 0.1302 |
| | | 375 | **0.0290** | **0.1150** | 0.0301 | 0.1186 | 0.0400 | 0.1296 | 0.0409 | 0.1335 | 0.0300 | 0.1186 | 0.0371 | 0.1246 | 0.0440 | 0.1379 | 0.0441 | 0.1424 |
| | W | 125 | **0.0093** | **0.0490** | 0.0102 | 0.0590 | 0.0113 | 0.0616 | 0.0404 | 0.0941 | 0.0111 | 0.0612 | 0.0123 | 0.0610 | 0.0124 | 0.0623 | 0.0682 | 0.1127 |
| | | 250 | **0.0163** | **0.0728** | 0.0166 | 0.0729 | 0.0351 | 0.1033 | 0.0995 | 0.1441 | 0.0173 | 0.0742 | 0.0364 | 0.0981 | 0.0336 | 0.0947 | 0.0881 | 0.1381 |
| | | 375 | **0.0193** | **0.0778** | 0.0207 | 0.0798 | 0.0470 | 0.1188 | 0.1135 | 0.1548 | 0.0206 | 0.0804 | 0.0483 | 0.1126 | 0.0435 | 0.1080 | 0.1219 | 0.1617 |
| | WT | 125 | **0.0129** | **0.0667** | 0.0135 | 0.0676 | 0.0195 | 0.0779 | 0.0646 | 0.1244 | 0.0145 | 0.0693 | 0.0180 | 0.0719 | 0.0216 | 0.0787 | 0.0733 | 0.1301 |
| | | 250 | **0.0206** | **0.0815** | 0.0217 | 0.0830 | 0.0449 | 0.1147 | 0.1127 | 0.1623 | 0.0211 | 0.0821 | 0.0437 | 0.1083 | 0.0425 | 0.1058 | 0.1041 | 0.1566 |
| | | 375 | **0.0233** | **0.0873** | 0.0248 | 0.0886 | 0.0552 | 0.1255 | 0.1149 | 0.1641 | 0.0245 | 0.0887 | 0.0474 | 0.1146 | 0.0456 | 0.1122 | 0.1165 | 0.1654 |

### 5.2.2 RESULTS AND DISCUSSION

**Quantitative Results:** Experiment results of the real-world datasets are shown in Table 3. We also conduct the Wilcoxon signed-rank test Richard (2021) on the reported accuracies, our method significantly outperforms the baselines, with a p-value threshold of 0.05. According to the experiment results, our **IDOL** model significantly outperforms all other baselines on most of the human motion forecasting tasks. Specifically, our method outperforms the most competitive baseline by a clear margin of 4%-34% and promotes the forecasting accuracy substantially on complex motions like the SittingDown (SD) and the Walking (W). This is because the human motion datasets contain more complex patterns described by stable causal structures, and our method can learn the latent causal process with identifiability guarantees. It is noted that our method achieves a better performance than that of TDRL, which does not consider the instantaneous dependencies among latent variables. We also find that our model achieves a comparable performance in the motion of CatchThrow (CT), this is because the size of this dataset is too small for the model to model the distribution of the observations. We further consider a more complex synthetic mixture of human motion forecasting, experiment results are shown in Appendix F.2.2. Please refer to Appendix F.2.4 for the experiment results of ablation studies. Moreover, we also investigate the proposed IDOL model on other high-dimension datasets in Appendix H.1.

## 6 CONCLUSION

This paper proposes a general framework for time series data with instantaneous dependencies to identify the latent variables and latent causal relations up to the Markov equivalence class. Furthermore, with mild assumption, the causal graph is also identifiable. Different from existing methods that require the assumptions of grouping observations and interventions, the proposed **IDOL** model employs the sparse latent process assumption, which is easy to satisfy in real-world time series data. We also devise a variational-inference-based method with sparsity regularization to build the gap between theories and practice. Experiment results on simulation datasets evaluate the effectiveness of latent variables identification and latent directed acyclic graphs reconstruction. Evaluation in human motion datasets with instantaneous dependencies reflects the practicability in real-world scenarios. There are two main limitations in our work. First, our method is not for high-dimensional time series data because modeling the complex dependencies through sparsity constraints is challenging. Second, our method relies on the invertible mixing process, but it may not hold in real-world scenarios. How to address these limitations and make our method more scalable will be an interesting direction.

## 7 ACKNOWLEDGMENT

We would like to acknowledge the support from NSF Award No. 2229881, AI Institute for Societal Decision Making (AI-SDM), the National Institutes of Health (NIH) under Contract R01HL159805, and grants from Quris AI, Florin Court Capital, and MBZUAI-WIS Joint Program. Moreover, this research was supported in part by the National Science and Technology Major Project (2021ZD0111501), Natural Science Foundation of China (U24A20233).

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

*Supplement to*

# "On the Identification of Temporally Causal Representation with Instantaneous Dependence"

Appendix organization:

# A    RELATED WORKS

## A.1    IDENTIFIABILITY OF CAUSAL REPRESENTATION LEARNING

To achieve the causal representation (Rajendran et al., 2024; Mansouri et al., 2023; Wendong et al., 2024) for time series data, several researchers leverage the independent component analysis (ICA) to recover the latent variables with identification guarantees (Yao et al., 2023; Schölkopf et al., 2021; Liu et al., 2023; Gresele et al., 2020). Conventional methods assume a linear mixing function from the latent variables to the observed variables (Comon, 1994; Hyvärinen, 2013; Lee & Lee, 1998; Zhang & Chan, 2007). To relax the linear assumption, researchers achieve the identifiability of latent variables via nonlinear ICA by using different types of assumptions like auxiliary variables or sparse generation process (Zheng et al., 2022; Hyvärinen & Pajunen, 1999; Hyvärinen et al., 2023; Khemakhem et al., 2020b; Li et al., 2023c). As for the methods that employ the auxiliary variables, Khemakhem et al. (2020a) first achieve the identifiability by assuming the latent sources with exponential family and introducing auxiliary variables e.g., domain indexes, time indexes, and class labels (Khemakhem et al., 2020a; Hyvarinen & Morioka, 2016; 2017; Hyvarinen et al., 2019). To further relax the exponential family assumption, Kong et al. (2022); Xie et al. (2022b); Kong et al. (2023); Yan et al. (2023); Xie et al. (2022b) achieve the component-wise identification results for nonlinear ICA with a $2n + 1$ number of auxiliary variables for $n$ latent variables. Recently, Li et al. (2024) further relax to $n + 1$ number of auxiliary variables and achieve subspace identifiability. To seek identifiability in an unsupervised manner, researchers employ the assumption of structural sparsity to achieve identifiability (Ng et al., 2024; Lachapelle et al., 2022; Zheng et al., 2022; Xu et al., 2024a). Specifically, Lachapelle et al. (2023); Lachapelle & Lacoste-Julien (2022) proposed mechanism sparsity regularization as an inductive bias to identify the causal latent factors and achieve the identifiability on several scenarios like multi-task learning. They further show the identifiability results up to the consistency relationship (Lachapelle et al., 2024a), which allows the partial disentanglement of latent variables. Recently, (Zhang et al., 2024) use the sparse structures of latent variables to achieve identifiability under distribution shift. Different from these methods that assume sparsity exists in the generation process between the latent source and the observation, we assume the *sparse latent process*, where sparsity exists in the transition of latent variables and is common in real-world time series data.

## A.2    NONLINEAR ICA FOR TIME SERIES DATA

Our work is also related to the temporally causal representation. Existing methods for temporally causal representation (Yan et al., 2023; Huang et al., 2023; H"alv"a & Hyvarinen, 2020; Lippe et al., 2022) usually rely on the conditional independence assumption, where the latent variables are mutually independent conditional on their time-delayed parents. Specifically, Hyvarinen & Morioka (2016) leverage the independent sources principle and the variability of data segments to achieve identifiability on nonstationary time series data. They further use permutation-based contrastive (Hyvarinen & Morioka, 2017) to achieve identifiability on stationary time series data. The recent advancements in temporally causal representation include LEAP (Yao et al., 2021), TDRL (Yao et al., 2022), which leverage the properties of independent noises and variability historical information. However, this assumption is too strong to meet in real-world scenarios since instantaneous dependencies are common in real-world scenarios like human motion. To solve this problem, researchers introduce further assumptions. For example, Lippe et al. (2023) propose the i-Citris, which demonstrates identifiability by assuming interventions on latent variables. In addition, Morioka & Hyvärinen (2023) propose G-CaRL, which yields identifiability with the assumption of the grouping of observational variables. However, these assumptions are still hard to meet in practice. Hence we propose a more relaxed assumption, transition sparsity, to achieve identifiability under time series data with instantaneous

## A.3    CAUSAL DISCOVERY WITH LATENT VARIABLES

Several studies are proposed to discover causally related latent variables (Chen et al., 2024b; Lachapelle et al., 2024b; Monti et al., 2020; Zhang & Hyvarinen, 2012; Tashiro et al., 2012). Specifically, (Huang et al., 2022; Kong et al., 2024) leverage the vanishing Tetrad conditions or rank constraints to identify latent variables in linear-Gaussian models. Xie et al. (2022a); Cai et al. (2019) further draw upon non-Gaussianity in their analysis for linear, non-Gaussian scenarios. Other

methods aim to reconstruct the hierarchical structures of the latent variables, like Xie et al. (2022a); Huang et al. (2022). However, these methods usually use the linear assumption and can hardly handle the real-world time series data with complex nonlinear relationships.

### A.4 Instantaneous Dependencey of Time Series Data

When the discrete time series data is sampled in a low-frequency or low-resolution manner, instantaneous dependence (Adak, 1996; Koutlis et al., 2019; Gersch, 1985; Jamaludeen et al., 2022; Wu et al., 2023a) occurs where variables at each time step are not independent given their historical observations. Swanson & Granger (1997) first discuss the instantaneous dependence from the perspective of noise. Lots of works investigate instantaneous dependency from the causal view. Specifically, Hyvärinen et al. (2010) estimate both instantaneous and lagged effects via a non-Gaussian model. Gong et al. (2015; 2017) discover causal structure with instantaneous dependency from subsampled and aggregated data. Recently, Zhu et al. (2023) consider instantaneous dependency in reinforcement learning.

### A.5 Time Series Forecasting

Time series forecasting (Hyndman & Athanasopoulos, 2018; Box & Pierce, 1970) is one of the most popular research problems in recent years. Recently, the deep-learning-based methods have made great progress, which can be categorized according to different types of neural architectures like the RNN-based methods (Hochreiter & Schmidhuber, 1997; Lai et al., 2018; Salinas et al., 2020), CNN-based models (Bai et al., 2018; Wang et al., 2022; Wu et al., 2022), and the methods based on state-space model (Gu et al., 2022; 2021b;a). Recently, Transformer-based methods (Zhou et al., 2021; Wu et al., 2021; Nie et al., 2022) further push the development of time series forecasting. However, these methods seldom consider the instantaneous dependencies of time series data.

## B Proof of Theory

### B.1 Relationships between Ground-truth and Estimated Latent Variables

**Theorem A1.** *(Identifiability of Temporally Latent Process) For a series of observations $\mathbf{x}_t \in \mathbb{R}^n$ and estimated latent variables $\hat{\mathbf{z}}_t \in \mathbb{R}^n$ with the corresponding process $\hat{f}_i, \hat{p}(\epsilon), \hat{g}$, where $\hat{g}$ is invertible, suppose the process subject to observational equivalence $\mathbf{x}_t = \hat{\mathbf{g}}(\hat{\mathbf{z}}_t)$. Let $\mathbf{c}_t \triangleq \{\mathbf{z}_{t-1}, \mathbf{z}_t\} \in \mathbb{R}^{2n}$ and $\mathcal{M}_{\mathbf{c}_t}$ be the variable set of two consecutive timestamps and the corresponding Markov network respectively. Suppose the following assumptions hold:*

- *A1 (Smooth and Positive Density): The conditional probability function of the latent variables $\mathbf{c}_t$ is smooth and positive, i.e., $p(\mathbf{c}_t|\mathbf{z}_{t-2})$ is third-order differentiable and $p(\mathbf{c}_t|\mathbf{z}_{t-2}) > 0$ over $\mathbb{R}^{2n}$,*

- *A2 (Sufficient Variability): Denote $|\mathcal{M}_{\mathbf{c}_t}|$ as the number of edges in Markov network $\mathcal{M}_{\mathbf{c}_t}$. Let*

$$
\begin{aligned}
w(m) = & \Big( \frac{\partial^3 \log p(\mathbf{c}_t|\mathbf{z}_{t-2})}{\partial c_{t,1}^2 \partial z_{t-2,m}}, \cdots, \frac{\partial^3 \log p(\mathbf{c}_t|\mathbf{z}_{t-2})}{\partial c_{t,2n}^2 \partial z_{t-2,m}} \Big) \oplus \\
& \Big( \frac{\partial^2 \log p(\mathbf{c}_t|\mathbf{z}_{t-2})}{\partial c_{t,1} \partial z_{t-2,m}}, \cdots, \frac{\partial^2 \log p(\mathbf{c}_t|\mathbf{z}_{t-2})}{\partial c_{t,2n} \partial z_{t-2,m}} \Big) \oplus \Big( \frac{\partial^3 \log p(\mathbf{c}_t|\mathbf{z}_{t-2})}{\partial c_{t,i} \partial c_{t,j} \partial z_{t-2,m}} \Big)_{(i,j) \in \mathcal{E}(\mathcal{M}_{\mathbf{c}_t})},
\end{aligned}
\tag{A1}
$$

*where $\oplus$ denotes concatenation operation and $(i,j) \in \mathcal{E}(\mathcal{M}_{\mathbf{c}_t})$ denotes all pairwise indice such that $c_{t,i}, c_{t,j}$ are adjacent in $\mathcal{M}_{\mathbf{c}_t}$. For $m \in [1, \cdots, n]$, there exist $4n + |\mathcal{M}_{\mathbf{c}_t}|$ different values of $\mathbf{z}_{t-2,m}$, such that the $4n + |\mathcal{M}_{\mathbf{c}_t}|$ values of vector functions $w(m)$ are linearly independent.*

*Then for any two different entries $\hat{c}_{t,k}, \hat{c}_{t,l}$ of $\hat{\mathbf{c}}_t \in \mathbb{R}^{2n}$ that are **not adjacent** in the Markov network $\mathcal{M}_{\hat{\mathbf{c}}_t}$ over estimated $\hat{\mathbf{c}}_t$,*
*(i) Each ground-truth latent variable $c_{t,i}$ of $\mathbf{c}_t \in \mathbb{R}^{2n}$ is a function of at most one of $\hat{c}_k$ and $\hat{c}_l$,*
*(ii) For each pair of ground-truth latent variables $c_{t,i}$ and $c_{t,j}$ of $\mathbf{c}_t \in \mathbb{R}^{2n}$ that are **adjacent** in $\mathcal{M}_{\mathbf{c}_t}$ over $\mathbf{c}_t$, they can not be a function of $\hat{c}_{t,k}$ and $\hat{c}_{t,l}$ respectively.*

*Proof.* We start from the matched marginal distribution to develop the relationship between $\mathbf{z}_t$ and $\hat{\mathbf{z}}_t$ as follows:

$$p(\hat{\mathbf{x}}_t) = p(\mathbf{x}_t) \Longleftrightarrow p(\hat{g}(\hat{\mathbf{z}}_t)) = p(g(z_t)) \Longleftrightarrow p((g^{-1} \circ \hat{g})(\hat{\mathbf{z}}_t)) = p(\mathbf{z}_t) \Longleftrightarrow p(h_z(\hat{\mathbf{z}}_t)) = p(\mathbf{z}_t), \quad \text{(A2)}$$

where $\hat{g} : \mathcal{Z} \to \mathcal{X}$ denotes the estimated mixing function, and $h := g^{-1} \circ \hat{g}$ is the transformation between the ground-truth latent variables and the estimated ones. Since $\hat{g}$ and $g$ are invertible, $h$ is invertible as well. Since Equation (A2) holds true for all time steps, there must exist an invertible function $h_c$ such that $p(h_c(\hat{\mathbf{c}}_t)) = p(\mathbf{c}_t)$, whose Jacobian matrix at time step $t$ is

$$\mathbf{J}_{h_c,t} = \begin{bmatrix} \mathbf{J}_{h_z,t-1} & 0 \\ 0 & \mathbf{J}_{h_z,t} \end{bmatrix}. \quad \text{(A3)}$$

Then for each value of $\mathbf{x}_{t-2}$, the Jacobian matrix of the mapping from $(\mathbf{x}_{t-2}, \hat{\mathbf{c}}_t)$ to $(\mathbf{x}_{t-2}, \mathbf{c}_t)$ can be written as follows:

$$\begin{bmatrix} \mathbf{I} & \mathbf{0} \\ * & \mathbf{J}_{h_c,t} \end{bmatrix},$$

where $*$ denotes any matrix. Since $\mathbf{x}_{t-2}$ can be fully characterized by itself, the left top and right top block are $\mathbf{1}$ and $\mathbf{0}$ respectively, and the determinant of this Jacobian matrix is the same as $|\mathbf{J}_{h_c,t}|$. Therefore, we have:

$$p(\hat{\mathbf{c}}_t, \mathbf{x}_{t-2}) = p(\mathbf{c}_t, \mathbf{x}_{t-2})|\mathbf{J}_{h_c,t}|. \quad \text{(A4)}$$

Dividing both sides of Equation (A4) by $p(\mathbf{x}_{t-2})$, we further have:

$$p(\hat{\mathbf{c}}_t|\mathbf{x}_{t-2}) = p(\mathbf{c}_t|\mathbf{x}_{t-2})|\mathbf{J}_{h_c,t}|. \quad \text{(A5)}$$

Since $p(\mathbf{c}_t|\mathbf{x}_{t-2}) = p(\mathbf{c}_t|g(\mathbf{z}_{t-2})) = p(\mathbf{c}_t|\mathbf{z}_{t-2})$, and similarly $p(\hat{\mathbf{c}}_t|\mathbf{x}_{t-2}) = p(\hat{\mathbf{c}}_t|\hat{\mathbf{z}}_{t-2})$, we have:

$$\log p(\hat{\mathbf{c}}_t|\hat{\mathbf{z}}_{t-2}) = \log p(\mathbf{c}_t|\mathbf{z}_{t-2}) + \log |\mathbf{J}_{h_c,t}|. \quad \text{(A6)}$$

Let $\hat{c}_{t,k}, \hat{c}_{t,l}$ be two different variables that are not adjacent in the estimated Markov network $\mathcal{M}_{\hat{\mathbf{c}}_t}$ over $\hat{\mathbf{c}}_t = \{\hat{\mathbf{z}}_{t-1}, \hat{\mathbf{z}}_t\}$. We conduct the first-order derivative w.r.t. $\hat{c}_{t,k}$ and have

$$\frac{\partial \log p(\hat{\mathbf{c}}_t|\hat{\mathbf{z}}_{t-2})}{\partial \hat{c}_{t,k}} = \sum_{i=1}^{2n} \frac{\partial \log p(\mathbf{c}_t|\mathbf{z}_{t-2})}{\partial c_{t,i}} \cdot \frac{\partial c_{t,i}}{\partial \hat{c}_{t,k}} + \frac{\partial \log |\mathbf{J}_{h_c,t}|}{\partial \hat{c}_{t,k}}. \quad \text{(A7)}$$

We further conduct the second-order derivative w.r.t. $\hat{c}_{t,k}$ and $\hat{c}_{t,l}$, then we have:

$$\begin{aligned}
\frac{\partial^2 \log p(\hat{\mathbf{c}}_t|\hat{\mathbf{z}}_{t-2})}{\partial \hat{c}_{t,k} \partial \hat{c}_{t,l}} = &\sum_{i=1}^{2n} \sum_{j=1}^{2n} \frac{\partial^2 \log p(\mathbf{c}_t|\mathbf{z}_{t-2})}{\partial c_{t,i} \partial c_{t,j}} \cdot \frac{\partial c_{t,i}}{\partial \hat{c}_{t,k}} \cdot \frac{\partial c_{t,j}}{\partial \hat{c}_{t,l}} \\
&+ \sum_{i=1}^{2n} \frac{\partial \log p(\mathbf{c}_t|\mathbf{z}_{t-2})}{\partial c_{t,i}} \cdot \frac{\partial^2 c_{t,i}}{\partial \hat{c}_{t,k} \partial \hat{c}_{t,l}} + \frac{\partial^2 \log |\mathbf{J}_{h_c,t}|}{\partial \hat{c}_{t,k} \partial \hat{c}_{t,l}}.
\end{aligned} \quad \text{(A8)}$$

Since $\hat{c}_{t,k}, \hat{c}_{t,l}$ are not adjacent in $\mathcal{M}_{\hat{\mathbf{c}}_t}$, $\hat{c}_{t,k}$ and $\hat{c}_{t,l}$ are conditionally independent given $\hat{\mathbf{c}}_t \backslash \{\hat{c}_{t,k}, \hat{c}_{t,l}\}$. Utilizing the fact that conditional independence can lead to zero cross derivative (Lin, 1997), for each value of $\hat{\mathbf{z}}_{t-2}$, we have

$$\begin{aligned}
\frac{\partial^2 \log p(\hat{\mathbf{c}}_t|\hat{\mathbf{z}}_{t-2})}{\partial \hat{c}_{t,k} \partial \hat{c}_{t,l}} = &\frac{\partial^2 \log p(\hat{c}_{t,k}|\hat{\mathbf{c}}_t \backslash \{\hat{c}_{t,k}, \hat{c}_{t,l}\}, \hat{\mathbf{z}}_{t-2})}{\partial \hat{c}_{t,k} \partial \hat{c}_{t,l}} + \frac{\partial^2 \log p(\hat{c}_{t,l}|\hat{\mathbf{c}}_t \backslash \{\hat{c}_{t,k}, \hat{c}_{t,l}\}, \hat{\mathbf{z}}_{t-2})}{\partial \hat{c}_{t,k} \partial \hat{c}_{t,l}} \\
&+ \frac{\partial^2 \log p(\hat{\mathbf{c}}_t \backslash \{\hat{c}_{t,k}, \hat{c}_{t,l}\}|\hat{\mathbf{z}}_{t-2})}{\partial \hat{c}_{t,k} \partial \hat{c}_{t,l}} = 0.
\end{aligned} \quad \text{(A9)}$$

Bring in Equation (A9), Equation (A8) can be further derived as

$$0 = \underbrace{\sum_{i=1}^{2n} \frac{\partial^2 \log p(\mathbf{c}_t|\mathbf{z}_{t-2})}{\partial c_{t,i}^2} \cdot \frac{\partial c_{t,i}}{\partial \hat{c}_{t,k}} \cdot \frac{\partial c_{t,i}}{\partial \hat{c}_{t,l}}}_{\textbf{(i) } i=j} + \underbrace{\sum_{i=1}^{2n} \sum_{j:(j,i)\in\mathcal{E}(\mathcal{M}_{\mathbf{c}_t})} \frac{\partial^2 \log p(\mathbf{c}_t|\mathbf{z}_{t-2})}{\partial c_{t,i}\partial c_{t,j}} \cdot \frac{\partial c_{t,i}}{\partial \hat{c}_{t,k}} \cdot \frac{\partial c_{t,j}}{\partial \hat{c}_{t,l}}}_{\textbf{(ii)} c_{t,i} \text{ and } c_{t,j} \text{ are adjacent in } \mathcal{M}_{\mathbf{c}_t}}$$

$$+ \underbrace{\sum_{i=1}^{2n} \sum_{j:(j,i)\notin\mathcal{E}(\mathcal{M}_{\mathbf{c}_t})} \frac{\partial^2 \log p(\mathbf{c}_t|\mathbf{z}_{t-2})}{\partial c_{t,i}\partial c_{t,j}} \cdot \frac{\partial c_{t,i}}{\partial \hat{c}_{t,k}} \cdot \frac{\partial c_{t,j}}{\partial \hat{c}_{t,l}}}_{\textbf{(iii)} c_{t,i} \text{ and } c_{t,j} \text{ are } \textbf{not} \text{ adjacent in } \mathcal{M}_{\mathbf{c}_t}}$$

$$+ \sum_{i=1}^{2n} \frac{\partial \log p(\mathbf{c}_t|\mathbf{z}_{t-2})}{\partial c_{t,i}} \cdot \frac{\partial^2 c_{t,i}}{\partial \hat{c}_{t,k}\partial \hat{c}_{t,l}} + \frac{\partial \log |\mathbf{J}_{h_c,t}|}{\partial \hat{c}_{t,k}\partial \hat{c}_{t,l}},$$

$$(A10)$$

where $(j,i) \in \mathcal{E}(\mathcal{M}_{\mathbf{c}_t})$ denotes that $c_{t,i}$ and $c_{t,j}$ are adjacent in $\mathcal{M}_{\mathbf{c}_t}$. Similar to Equation (A9), we have $\frac{\partial^2 p(\mathbf{c}_t|\mathbf{z}_{t-2})}{\partial c_{t,i}\partial c_{t,j}} = 0$ when $c_{t,i}, c_{t,j}$ are not adjacent in $\mathcal{M}_{\mathbf{c}_t}$. Thus, Equation (A10) can be rewritten as

$$0 = \sum_{i=1}^{2n} \frac{\partial^2 \log p(\mathbf{c}_t|\mathbf{z}_{t-2})}{\partial c_{t,i}^2} \cdot \frac{\partial c_{t,i}}{\partial \hat{c}_{t,k}} \cdot \frac{\partial c_{t,i}}{\partial \hat{c}_{t,l}} + \sum_{i=1}^{2n} \sum_{j:(j,i)\in\mathcal{E}(\mathcal{M}_{\mathbf{c}})} \frac{\partial^2 \log p(\mathbf{c}_t|\mathbf{z}_{t-2})}{\partial c_{t,i}\partial c_{t,j}} \cdot \frac{\partial c_{t,i}}{\partial \hat{c}_{t,k}} \cdot \frac{\partial c_{t,j}}{\partial \hat{c}_{t,l}}$$

$$+ \sum_{i=1}^{2n} \frac{\partial \log p(\mathbf{c}_t|\mathbf{z}_{t-2})}{\partial c_{t,i}} \cdot \frac{\partial^2 c_{t,i}}{\partial \hat{c}_{t,k}\partial \hat{c}_{t,l}} + \frac{\partial \log |\mathbf{J}_{h_c,t}|}{\partial \hat{c}_{t,k}\partial \hat{c}_{t,l}}.$$

$$(A11)$$

Then for each $m = 1, 2, \cdots, n$ and each value of $z_{t-2,m}$, we conduct partial derivative on both sides of Equation (A11) and have:

$$0 = \sum_{i=1}^{2n} \frac{\partial^3 \log p(\mathbf{c}_t|\mathbf{z}_{t-2})}{\partial c_{t,i}^2 \partial z_{t-2,m}} \cdot \frac{\partial c_{t,i}}{\partial \hat{c}_{t,k}} \cdot \frac{\partial c_{t,i}}{\partial \hat{c}_{t,l}} + \sum_{i=1}^{2n} \sum_{j:(j,i)\in\mathcal{E}(\mathcal{M}_{\mathbf{c}})} \frac{\partial^3 \log p(\mathbf{c}_t|\mathbf{z}_{t-2})}{\partial c_{t,i}\partial c_{t,j}\partial z_{t-2,m}} \cdot \frac{\partial c_{t,i}}{\partial \hat{c}_{t,k}} \cdot \frac{\partial c_{t,j}}{\partial \hat{c}_{t,l}},$$

$$+ \sum_{i=1}^{2n} \frac{\partial^2 \log p(c_t|\mathbf{z}_{t-2})}{\partial c_{t,i}\partial z_{t-2,m}} \cdot \frac{\partial c_{t,i}^2}{\partial \hat{c}_{t,k}\partial \hat{c}_{t,l}}$$

$$(A12)$$

Finally we have

$$0 = \sum_{i=1}^{2n} \frac{\partial^3 \log p(\mathbf{c}_t|\mathbf{z}_{t-2})}{\partial c_{t,i}^2 \partial z_{t-2,m}} \cdot \frac{\partial c_{t,i}}{\partial \hat{c}_{t,k}} \cdot \frac{\partial c_{t,i}}{\partial \hat{c}_{t,l}} + \sum_{i=1}^{2n} \frac{\partial^2 \log p(c_t|\mathbf{z}_{t-2})}{\partial c_{t,i}\partial z_{t-2,m}} \cdot \frac{\partial c_{t,i}^2}{\partial \hat{c}_{t,k}\partial \hat{c}_{t,l}}$$

$$+ \sum_{i,j:(j,i)\in\mathcal{E}(\mathcal{M}_{\mathbf{c}})} \frac{\partial^3 \log p(\mathbf{c}_t|\mathbf{z}_{t-2})}{\partial c_{t,i}\partial c_{t,j}\partial z_{t-2,m}} \cdot \left( \frac{\partial c_{t,i}}{\partial \hat{c}_{t,k}} \cdot \frac{\partial c_{t,j}}{\partial \hat{c}_{t,l}} + \frac{\partial c_{t,j}}{\partial \hat{c}_{t,k}} \cdot \frac{\partial c_{t,i}}{\partial \hat{c}_{t,l}} \right).$$

$$(A13)$$

According to Assumption A2, we can construct $4n + |\mathcal{M}_{\mathbf{c}}|$ different equations with different values of $z_{t-2,m}$, and the coefficients of the equation system they form are linearly independent. To ensure that the right-hand side of the equations are always 0, the only solution is

$$\frac{\partial c_{t,i}}{\partial \hat{c}_{t,k}} \cdot \frac{\partial c_{t,i}}{\partial \hat{c}_{t,l}} = 0, \tag{A14}$$

$$\frac{\partial c_{t,i}}{\partial \hat{c}_{t,k}} \cdot \frac{\partial c_{t,j}}{\partial \hat{c}_{t,l}} + \frac{\partial c_{t,j}}{\partial \hat{c}_{t,k}} \cdot \frac{\partial c_{t,i}}{\partial \hat{c}_{t,l}} = 0, \tag{A15}$$

$$\frac{\partial c_{t,i}^2}{\partial \hat{c}_{t,k}\partial \hat{c}_{t,l}} = 0. \tag{A16}$$

Bringing Eq A14 into Eq A15, at least one product must be zero, thus the other must be zero as well. That is,

$$\frac{\partial c_{t,i}}{\partial \hat{c}_{t,k}} \cdot \frac{\partial c_{t,j}}{\partial \hat{c}_{t,l}} = 0. \tag{A17}$$

According to the aforementioned results, for any two different entries $\hat{c}_{t,k}, \hat{c}_{t,l} \in \hat{\mathbf{c}}_t$ that are **not adjacent** in the Markov network $\mathcal{M}_{\hat{\mathbf{c}}_t}$ over estimated $\hat{\mathbf{c}}_t$, we draw the following conclusions.
**(i)** Equation (A14) implies that, each ground-truth latent variable $c_{t,i} \in \mathbf{c}_t$ is a function of at most one of $\hat{c}_{t,k}$ and $\hat{c}_{t,l}$,
**(ii)** Equation (A17) implies that, for each pair of ground-truth latent variables $c_{t,i}$ and $c_{t,j}$ that are **adjacent** in $\mathcal{M}_{\mathbf{c}_t}$ over $\mathbf{c}_t$, they can not be a function of $\hat{c}_{t,k}$ and $\hat{c}_{t,l}$ respectively. $\qquad\square$

### B.2 EXTENSION TO MULTIPLE LAGS AND SEQUENCE LENGTHS

For the sake of simplicity, we consider only one special case with $\tau = 1$ and $L = 2$ in Theorem A2. Our identifiability theorem can be actually extended to arbitrary lags and subsequences easily. For any given $\tau$, and subsequence which is centered at $\mathbf{z}_t$ with previous $lo$ and following $hi$ steps, i.e., $\mathbf{c}_t = \{\mathbf{z}_{t-lo}, \cdots, \mathbf{z}_t, \cdots, \mathbf{z}_{t+hi}\} \in \mathbb{R}^{(lo+hi+1) \times n}$. In this case, the vector function $w(i, j, m)$ in Sufficient Variability Assumption should be modified as

$$
\begin{aligned}
w(i,j,m) =& \left( \frac{\partial^3 \log p(\mathbf{c}_t | \mathbf{z}_{t-lo-1}, \cdots, \mathbf{z}_{t-lo-\tau})}{\partial c_{t,1}^2 \partial z_{t-lo-1,m}}, \cdots, \frac{\partial^3 \log p(\mathbf{c}_t | \mathbf{z}_{t-lo-1}, \cdots, \mathbf{z}_{t-lo-\tau})}{\partial c_{t,2n}^2 \partial z_{t-lo-1,m}} \right) \oplus \\
& \left( \frac{\partial^2 \log p(c_t | \mathbf{z}_{t-lo-1}, \cdots, \mathbf{z}_{t-lo-\tau})}{\partial c_{t,1} \partial z_{t-lo-1,m}}, \cdots, \frac{\partial^2 \log p(c_t | \mathbf{z}_{t-lo-1}, \cdots, \mathbf{z}_{t-lo-\tau})}{\partial c_{t,2n} \partial z_{t-lo-1,m}} \right) \oplus \\
& \left( \frac{\partial^3 \log p(\mathbf{c}_t | \mathbf{z}_{t-lo-1}, \cdots, \mathbf{z}_{t-lo-\tau})}{\partial c_{t,i} \partial c_{t,j} \partial z_{t-lo-1,m}} \right)_{(i,j) \in \mathcal{E}(\mathcal{M}_{\mathbf{c}_t})}.
\end{aligned} \tag{A18}
$$

Besides, $2 \times n \times (lo + hi + 1) + |\mathcal{M}_{\mathbf{c}_t}|$ values of linearly independent vector functions in $z_{t',m}$ for $t' \in [t - lo - 1, \cdots, t - lo - \tau]$ and $m \in [1, \cdots, n]$ are required as well. The rest part of the theorem remains the same, and the proof can be easily extended in such a setting.

### B.3 IDENTIFIABLITY OF LATENT VARIABLES

**Theorem A2.** *(Component-wise Identification of Latent Variables with instantaneous dependencies.)* *Suppose that the observations are generated by Equation (1)-(2), and $\mathcal{M}_{\mathbf{c}_t}$ is the Markov network over $\mathbf{c}_t = \{\mathbf{z}_{t-1}, \mathbf{z}_t, \mathbf{z}_{t+1}\} \in \mathbb{R}^{3n}$. Except for the assumptions A1 and A2 from Theorem 1, we further make the following assumption:*

- *A3 (Sparse Latent Process): For any $z_{t,i} \in \mathbf{z}_t$, the intimate neighbor set of $z_{t,i}$ is an empty set.*

*When the observational equivalence is achieved with the minimal number of edges of estimated Markov network of $\mathcal{M}_{\hat{\mathbf{c}}_t}$, then we have the following two statements:*

*(i) The estimated Markov network $\mathcal{M}_{\hat{\mathbf{c}}_t}$ is isomorphic to the ground-truth Markov network $\mathcal{M}_{\mathbf{c}_t}$.*

*(ii) There exists a permutation $\pi$ of the estimated latent variables, such that $z_{t,i}$ and $\hat{z}_{t,\pi(i)}$ is one-to-one corresponding, i.e., $z_{t,i}$ is component-wise identifiable.*

*Proof.* First, we demonstrate that there always exists a row permutation for each invertible matrix such that the permuted diagonal entries are non-zero (Zhang et al., 2024). By contradiction, if the product of the diagonal entry of an invertible matrix $A$ is zero for every row permutation, then we have Equation

$$\det(A) = \sum_{\sigma \in \mathcal{S}_n} \left( \text{sgn}(\sigma) \prod_{i=1}^n a_{\sigma(i),i} \right), \tag{A19}$$

by the Leibniz formula, where $\mathcal{S}_n$ is the set of $n$-permutations. Thus, we have

$$\prod_{i=1}^n a_{\sigma(i),i} = 0, \quad \forall \sigma \in \mathcal{S}_n, \tag{A20}$$

which indicates that $det(A) = 0$ and $A$ is non-invertible. It contradicts the assumption that $A$ is invertible, and a row permutation where the permuted diagonal entries are non-zero must exist. Since $h_z$ is invertible, for $\mathbf{z}_t$ at time step $t$, there exists a permuted version of the estimated latent variables, such that

$$\frac{\partial z_{t,i}}{\partial \hat{z}_{t,\pi_t(i)}} \neq 0, \quad i = 1, \cdots, n, \tag{A21}$$

where $\pi_t$ is the corresponding permutation at time step $t$. Since $\mathbf{c}_t = \{\mathbf{z}_{t-1}, \mathbf{z}_t, \mathbf{z}_{t+1}\}$, by applying $\pi_{t-1}, \pi_t, \pi_{t+1}$, we have $\pi'$ such that

$$\frac{\partial c_{t,i}}{\partial \hat{c}_{t,\pi'(i)}} \neq 0, \quad i = 1, \cdots, 3n. \tag{A22}$$

Second, we demonstrate that $\mathcal{M}_{\mathbf{c}_t}$ is identical to $\mathcal{M}_{\hat{\mathbf{c}}_t^{\pi'}}$, where $\mathcal{M}_{\hat{\mathbf{c}}_t^{\pi'}}$ denotes the Markov network of the permuted version of $\pi'(\hat{\mathbf{c}}_t)$.

On the one hand, for any pair of $(i, j)$ such that $c_{t,i}, c_{t,j}$ are **adjacent** in $\mathcal{M}_{\mathbf{c}_t}$ while $\hat{c}_{t,\pi'(i)}, \hat{c}_{t,\pi'(j)}$ are **not adjacent** in $\mathcal{M}_{\hat{\mathbf{c}}_t^{\pi'}}$, according to Equation (A17), we have $\frac{\partial c_{t,i}}{\partial \hat{c}_{t,\pi'(i)}} \cdot \frac{\partial c_{t,j}}{\partial \hat{c}_{t,\pi'(j)}} = 0$, which is a contradiction with how $\pi'$ is constructed. Thus, any edge presents in $\mathcal{M}_{\mathbf{c}_t}$ must exist in $\mathcal{M}_{\hat{\mathbf{c}}_t^{\pi'}}$. On the other hand, since observational equivalence can be achieved by the true latent process $(g, f, p_{\mathbf{c}_t})$, the true latent process is clearly the solution with minimal edges.

Under the sparsity constraint on the edges of $\mathcal{M}_{\hat{\mathbf{c}}_t^{\pi'}}$, the permuted estimated Markov network $\mathcal{M}_{\hat{\mathbf{c}}_t^{\pi'}}$ must be identical to the true Markov network $\mathcal{M}_{\mathbf{c}_t}$. Thus, we claim that

(i) the estimated Markov network $\mathcal{M}_{\hat{\mathbf{c}}_t}$ is isomorphic to the ground-truth Markov network $\mathcal{M}_{\mathbf{c}_t}$.

Sequentially, under the same permutation $\pi_t$, we further give the proof that $z_{t,i}$ is only the function of $\hat{z}_{t,\pi_t(i)}$. Since the permutation happens on each time step respectively, the cross-time disentanglement is prevented clearly.

Now let us focus on instantaneous disentanglement. Suppose there exists a pair of indices $i, j \in \{1, \cdots, n\}$. According to Equation (A21), we have $\frac{\partial z_{t,i}}{\partial \hat{z}_{t,\pi_t(i)}} = 0$ and $\frac{\partial z_{t,j}}{\partial \hat{z}_{t,\pi_t(j)}} = 0$. Let us discuss it case by case.

- If $z_{t,i}$ is not adjacent to $z_{t,j}$, we have $\hat{z}_{t,\pi_t(i)}$ is not adjacent to $\hat{z}_{t,\pi_t(j)}$ as well according to the conclusion of identical Markov network. Using Equation (A14), we have $\frac{\partial z_{t,i}}{\partial \hat{z}_{t,\pi_t(i)}} \cdot \frac{\partial z_{t,i}}{\partial \hat{z}_{t,\pi_t(j)}} = 0$, which leads to $\frac{\partial z_{t,i}}{\partial \hat{z}_{t,\pi_t(j)}} = 0$.

- If $z_{t,i}$ is adjacent to $z_{t,j}$, we have $\hat{z}_{t,\pi_t(i)}$ is adjacent to $\hat{z}_{t,\pi_t(j)}$. When the Assumption A3 (Sparse Latent Process) is assured, i.e., the intimate neighbor set of $z_{t,i}$ is empty, there exists at least one pair of $(t', k)$ such that $z_{t',k}$ is adjacent to $z_{t,i}$ but not adjacent to $z_{t,j}$. Similarly, we have the same structure on the estimated Markov network, which means that $\hat{z}_{t',\pi_{t'}(k)}$ is adjacent to $\hat{z}_{t,\pi_t(i)}$ but not adjacent to $\hat{z}_{t,\pi_t(j)}$. Using Equation (A17) we have $\frac{\partial z_{t,k}}{\partial \hat{z}_{t',\pi_{t'}(k)}} \cdot \frac{\partial z_{t,i}}{\partial \hat{z}_{t,\pi_t(j)}} = 0$, which leads to $\frac{\partial z_{t,i}}{\partial \hat{z}_{t,\pi_t(j)}} = 0$.

In conclusion, we always have $\frac{\partial z_{t,i}}{\partial \hat{z}_{t,\pi_t(j)}} = 0$. Thus, we have reached the conclusion that

(ii) there exists a permutation $\pi$ of the estimated latent variables, such that $z_{t,i}$ and $\hat{z}_{t,\pi(i)}$ is one-to-one corresponding, i.e., $z_{t,i}$ is component-wise identifiable.

$\square$

## B.4 GENERAL CASE FOR COMPONENT-WISE IDENTIFICATIONS

In this part, we briefly give the proof for a more general case of our theorem.

**Corollary A1.** *(General Case for Component-wise Identification.)* *Suppose that the observations are generated by Equation (1)-(2), and there exists* $\mathbf{c}_t = \{\mathbf{z}_{t-a}, \cdots, \mathbf{z}_t, \cdots, \mathbf{z}_{t+b}\} \in \mathbb{R}^{(a+b+1)\times n}$

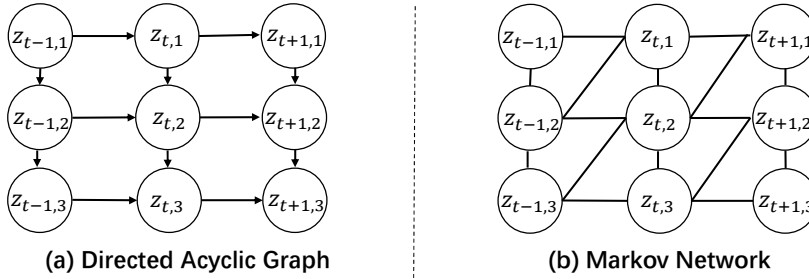

Figure A5: An example of DAG (a) and Markov Network (b).

*with the corresponding Markov network $\mathcal{M}_{\mathbf{c}_t}$. Suppose assumptions A1 and A2 hold true, and for any $z_{t,i} \in \mathbf{z}_t$, the intimate neighbor set of $z_{t,i}$ is an empty set. When the observational equivalence is achieved with the minimal number of edges of estimated Markov network of $\mathcal{M}_{\hat{\mathbf{c}}}$, there must exist a permutation $\pi$ of the estimated latent variables, such that $z_{t,i}$ and $\hat{z}_{t,\pi(i)}$ is one-to-one corresponding, i.e., $z_{t,i}$ is component-wise identifiable.*

*Proof.* The proof is similar to that of Theorem A2. The only difference is that given a different subsequence, the variables that are used to make intimate neighbors empty might be different. The rest part of the theorem remains the same. $\square$

Here we further discuss the idea behind the Sparse Latent Process. For two latent variables $z_{t,i}, z_{t,j}$ that are entangled at some certain timestamp, the contextual information can be utilized to recover these variables. Intuitively speaking, when $z_{t,i}$ is directly affected by some previous variable, says $z_{t-1,k}$, while $z_{t,j}$ is not. In this case, the changes that happen on $z_{t-1,k}$ can be captured, which helps to tell $z_{t,i}$ from $z_{t,j}$. Similarly, if $z_{t,i}$ directly affects $z_{t+1,k}$ while $z_{t,j}$ does not, we can distinguish $z_{t,i}$ from $z_{t,j}$ as well. When all variables are naturally conditionally independent, no contextual information will be needed. One more thing to note is that, even though the sparse latent process is not fully satisfied, as long as some structures mentioned above exist, the corresponding entanglement can be prevented.

### B.5 IDENTIFIABILITY OF LATENT CAUSAL PROCESS

Building on the results of Theorem 2, the latent variables are component-wise identifiable, which directly implies the identifiability of the latent causal process up to the Markov equivalence class. Leveraging temporal information allows one to further refine this identification beyond the equivalence class. If each latent variable $z_t$ has at least one temporal ancestor, one can establish full identifiability of the graph over the latent processes.

**Theorem A3.** *(Identification of Latent Causal Process.) Suppose that the observations are generated by Equation (1)-(2), and that $\mathcal{M}_{\mathbf{c}_t}$ is the Markov network over $\mathbf{c}_t = \{\mathbf{z}_{t-1}, \mathbf{z}_t\} \in \mathbb{R}^{2n}$. Suppose that all assumptions for Theorem 2 hold. We further make the following assumption: for any pair of adjacent latent variables $z_{t,i}, z_{t,j}$ at time step $t$, their time-delayed parents are not identical, i.e., $Pa_d(z_{t,i}) \neq Pa_d(z_{t,j})$. Then the causal graph of the latent causal process is identifiable.*

*Proof.* Since all assumptions for Theorem 2 hold, latent variable $\mathbf{z}_t$ is component-wise identifiable, i.e., there exists a permutation $\pi$ and invertible functions $h_i$ such that $z_{t,i} = h_i(\hat{z}_{t,\pi(i)})$. As $\mathbf{c}_t$ is nothing but a concatenation of $\mathbf{z}_t$ from different time steps, $\mathbf{c}_t$ is also component-wise identifiable with corresponding $\pi'$ and $h'_i$. Therefore, their conditional independence relationships are consistent:

$$c_{t,i} \perp c_{t,j} \mid \{c_{t,k} \mid k \in S\} \Rightarrow \hat{c}_{t,\pi'(i)} \perp \hat{c}_{t,\pi'(j)} \mid \{\hat{c}_{t,\pi'(k)} \mid k \in S\} \tag{A23}$$

for all $S \subseteq \{1, 2, \cdots, 2n\} \setminus \{i, j\}$. In this way, we transform the problem of causal discovery for latent variables into a problem of causal discovery for observable variables.

We first prove that the skeleton of the estimated causal graph is identical to the true skeleton.

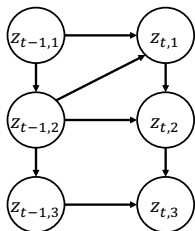

Figure A6: The example of dataset A whose causal graph is identifiable.

- On the one hand, if $c_{t,i}$ and $c_{t,j}$ are not adjacent, there exists a d-separation set $S_d$ such that $c_{t,i} \perp c_{t,j} \mid \{c_{t,k} \mid k \in S_d\}$. Meanwhile, we have $\hat{c}_{t,\pi'(i)} \perp \hat{c}_{t,\pi'(j)} \mid \{\hat{c}_{t,\pi'(k)} \mid k \in S_d\}$. Thus, $\hat{c}_{t,\pi'(i)}$ and $\hat{c}_{t,\pi'(j)}$ are not adjacent as well.

- On the other hand, if $c_{t,i}$ and $c_{t,j}$ are adjacent, there will be no d-separation set for $\hat{c}_{t,\pi'(i)}, \hat{c}_{t,\pi'(j)}$, meaning that $\hat{c}_{t,\pi'(i)}$ and $\hat{c}_{t,\pi'(j)}$ are adjacent in this case.

In conclusion, the skeleton is identifiable.

Then we prove that the time-delayed edges can be identified. Consider any $c_{t,i}$ from $\mathbf{z}_{t-1}$ and $c_{t,j}$ from $\mathbf{z}_t$. Similarly, we have $\hat{c}_{t,\pi'(i)}$ from $\hat{\mathbf{z}}_{t-1}$ and $\hat{c}_{t,\pi'(j)}$ from $\hat{\mathbf{z}}_t$. Since the skeleton is identified and the direction is implied by temporal information, specifically going from time step $t-1$ to $t$, there exists an edge from $\hat{c}_{t,\pi'(i)}$ to $\hat{c}_{t,\pi'(j)}$ if and only if $c_{t,i}$ points to $c_{t,j}$. Thus the temporally latent causal process can be identified.

Finally, we determine the direction of instantaneous edges. Since the skeleton is already identifiable, we only need to determine the direction. Consider any pair of $c_{t,i}$ and $c_{t,j}$ from $\mathbf{z}_t$, where $c_{t,i} \to c_{t,j}$. Since $\mathrm{Pa}_d(c_{t,i}) \neq \mathrm{Pa}_d(c_{t,j})$, there exists at least one $c_{t,k}$ from $\mathbf{z}_{t-1}$ such that $c_{t,k}$ points to exactly one of $c_{t,i}$ or $c_{t,j}$. We proceed with a case-by-case analysis.

- In case 1, we have $c_{t,k} \to c_{t,i} \to c_{t,j}$ where $c_{t,k}, c_{t,j}$ are not adjacent, and $c_{t,i}$ is in the d-seperation of $c_{t,k}$ and $c_{t,j}$. Meanwhile, in the estimated skeleton, $\hat{c}_{t,\pi'(i)}$ is also in the d-seperation of $\hat{c}_{t,\pi'(k)}$ and $\hat{c}_{t,\pi'(j)}$. Since the direction of time-delayed edge $\hat{c}_{t,\pi'(k)} \to \hat{c}_{t,\pi'(i)}$ is the known in the estimated skeleton $\hat{c}_{t,\pi'(k)} - \hat{c}_{t,\pi'(i)} - \hat{c}_{t,\pi'(j)}$, we deduce $\hat{c}_{t,\pi'(i)} \to \hat{c}_{t,\pi'(j)}$ based on the d-seperation.

- In case 2, we have $c_{t,i} \to c_{t,j} \leftarrow c_{t,k}$ where $c_{t,k}, c_{t,i}$ are not adjacent. Similarly, we have that $\hat{c}_{t,\pi'(j)}$ is not in the d-seperation of $\hat{c}_{t,\pi'(k)}$ and $\hat{c}_{t,\pi'(i)}$, and it can be shown that $\hat{c}_{t,\pi'(i)} \to \hat{c}_{t,\pi'(j)}$.

In conclusion, the identifiability of the instantaneous latent causal graph is established. Consequently, the entire latent causal process, encompassing both the time-delayed and instantaneous components, is identifiable up to the causal graph.

$\square$

**Discussion**: To further illustrate this Corollary, we use dataset A as an example, whose causal graph is shown in Figure A6. Since the skeleton and directions of time-delayed edges are straightforward to determine, we primarily focus on analyzing the directions of instantaneous edges within $\mathbf{z}_t$. Since $z_{t-1,3} \to z_{t,3} \leftarrow z_{t,2}$ is a v-structure, $z_{t,3} \leftarrow z_{t,2}$ can be determined. Since $z_{t-1,1} \to z_{t,1} \to z_{t,2}$ is a chain, $z_{t-1,1}$ and $z_{t,2}$ are not adjacent, and $z_{t-1,1} \to z_{t,1}$ is known, we have $z_{t,1} \to z_{t,2}$. Thus, the causal graph is identifiable. As shown in Figure 4, the model can learn the true causal graph.

### B.6 MARKOV NETWORK

A Markov network (or Markov random field) is a graphical model that represents the joint distribution of a set of random variables using an undirected graph.

**Definition 3** (Markov Network). *Markov network is an undirected graph $G = (V, E)$ with a set of random variables $X_{v \in V}$, where any two non-adjacent variables are conditionally independent given all other variables. That is,*

$$X_a \perp X_b | X_{V \setminus \{a,b\}}, \quad \forall (a,b) \notin E. \tag{A24}$$

Markov Networks and Directed Acyclic Graphs (DAGs) are both graphical models employed to represent joint distributions and to illustrate conditional independence properties. As shown in Figure A5, both of them are utilized to describe the latent causal process, yet they do not have to be equivalent.

### B.7 ISOMORPHISM OF MARKOV NETWORKS

**Definition 4** (Isomorphism of Markov networks). *We let the $V(\cdot)$ be the vertical set of any graphs, an isomorphism of Markov networks $M$ and $\hat{M}$ is a bijection between the vertex sets of $M$ and $\hat{M}$*

$$f : V(M) \to V(\hat{M})$$

*such that any two vertices $u$ and $v$ of $M$ are adjacent in $G$ if and only if $f(u)$ and $f(v)$ are adjacent in $\hat{M}$.*

### B.8 ILLUSTRATION OF INTIMATE NEIGHBOR SET

Here, we provide an example for a better understanding of the Intimate Neighbor Set. Take Figure A5(b) as an example. If we consider only one time step, says, $\mathbf{z}_t$, $\{z_{t,1}, z_{t,2}, z_{t,3}\}$ forms a clique. Thus we have $\Psi(z_{t,1}) = \{z_{t,2}, z_{t,3}\}$, since $z_{t,2}$ is adjacent to $z_{t,1}$ and all other neighbours of $z_{t,1}$, i.e., $z_{t,3}$. Similarily, we have $\Psi(z_{t,1}) = \{z_{t,1}, z_{t,3}\}$, $\Psi(z_{t,1}) = \{z_{t,1}, z_{t,2}\}$ as well. In this case, none of them is identifiable. In contrast, if we take all 3 time steps into consideration, the Intimate set of all latent varibles becomes empty. For example, $z_{t,2}$ is adjacent to $z_{t,1}$ but not adjacent to at least one neighbour of $z_{t,1}$, i.e., $z_{t-1,1}$, thus $z_{t,2} \notin \Psi(z_{t,1})$. Similarly, do this test for all other pairs variables, and the conclusion can be achieved.

## C DISCUSSION OF ASSUMPTIONS

To enhance understanding of our theoretical results, we provide some explanations of the assumptions, their connections to real-world scenarios, as well as the potential boundary of theoretical results.

First, The smooth and positive density assumption is standard in the literature on nonlinear ICA (Khemakhem et al., 2020a; Yao et al., 2022; Kong et al., 2022), meaning that the latent variables $\mathbf{z}_t$ change continuously based on historical information. For instance, in a weather dataset, temperature varies smoothly over time. However, this assumption may be violated if we cannot fully capture the transition probabilities from the observations. To mitigate this, we can sample a larger dataset to better estimate the latent causal process.

Second, the sufficient variability assumption has also been commonly adopted for the existing identifiable results of temporally causal representation learning (Yao et al., 2021; 2022; Chen et al., 2024a). This assumption describes the changeability of latent variables. Take human motion forecasting as an example, the latent variable may represent joint location at different time steps. The linear independence of the latent variables means that the changes of each joint cannot be linearly represented by others. Besides, while the sufficiency assumption is foundational to the theory of identifiability, it is not excessively restrictive. Even in cases where this assumption is not fully satisfied, it is still possible to achieve a degree of subspace identifiability (Kong et al., 2022).

Finally, the sparse latent process is the key assumption of our method, which is common in real-world scenarios. Intuitively, in human motion forecasting, the joints of humans can be considered as latent variables. Since there are few connections among the joints of the human body, the latent process described by the skeleton motion trajectory is sparse. Even though the sparsity assumption is not completely satisfied, we can still attain a subspace level of identifiability (Li et al., 2024). In this case, each true variable can be a function of, at most, an estimated version of its corresponding variable and those within the intimate set. Let us provide a simple example here. In a video of a moving car, it

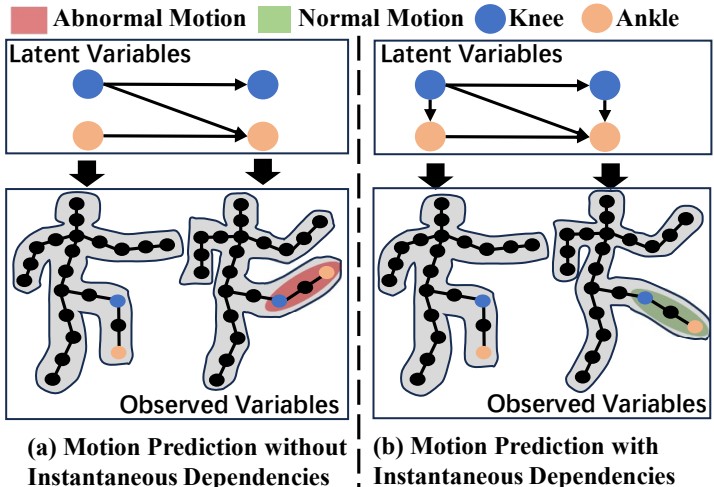

Figure A7: An example of human motion forecasting, where joints can be considered as latent variables and the latent skeleton decides the motion.

might be hard to have individual identifiability of the separate car wheels and car body; however, they can be considered as essential parts of the macro variable 'car'. This macro representation might be sufficient for the purpose of modeling the interactions between the car and other objects.

## D    THE IMPORTANCE OF CONSIDERING INSTANTANEOUS DEPENDENCE

Here, we provide an example as shown in Figure A7 to show why instantaneous dependence is important in time-series modeling. As shown in Figure A7 (a), without considering the instantaneous dependencies of a knee and an ankle, an abnormal motion might be generated, where the leg is bent at a distorted angle. Meanwhile, by taking the instantaneous dependencies into account as shown in Figure A7 (b), the predicted motion complies with human physiological structure.

## E    IMPLEMENTATION DETAILS

### E.1    PRIOR LIKELIHOOD DERIVATION

We first consider the prior of $\ln p(\mathbf{z}_{1:t})$. We start with an illustrative example of stationary latent causal processes with two time-delay latent variables, i.e. $\mathbf{z}_t = [z_{t,1}, z_{t,2}]$ with maximum time lag $L = 1$, i.e., $z_{t,i} = f_i(\mathbf{z}_{t-1}, \epsilon_{t,i})$ with mutually independent noises. Then we write this latent process as a transformation map $\mathbf{f}$ (note that we overload the notation $f$ for transition functions and for the transformation map):

$$\begin{bmatrix} z_{t-1,1} \\ z_{t-1,2} \\ z_{t,1} \\ z_{t,2} \end{bmatrix} = \mathbf{f}\left(\begin{bmatrix} z_{t-1,1} \\ z_{t-1,2} \\ \epsilon_{t,1} \\ \epsilon_{t,2} \end{bmatrix}\right).$$

By applying the change of variables formula to the map $\mathbf{f}$, we can evaluate the joint distribution of the latent variables $p(z_{t-1,1}, z_{t-1,2}, z_{t,1}, z_{t,2})$ as

$$p(z_{t-1,1}, z_{t-1,2}, z_{t,1}, z_{t,2}) = \frac{p(z_{t-1,1}, z_{t-1,2}, \epsilon_{t,1}, \epsilon_{t,2})}{|\det \mathbf{J_f}|}, \tag{A25}$$

where $\mathbf{J_f}$ is the Jacobian matrix of the map $\mathbf{f}$, where the instantaneous dependencies are assumed to be a low-triangular matrix:

$$\mathbf{J_f} = \begin{bmatrix} 1 & 0 & 0 & 0 \\ 0 & 1 & 0 & 0 \\ \frac{\partial z_{t,1}}{\partial z_{t-1,1}} & \frac{\partial z_{t,1}}{\partial z_{t-1,2}} & \frac{\partial z_{t,1}}{\partial \epsilon_{t,1}} & 0 \\ \frac{\partial z_{t,2}}{\partial z_{t-1,1}} & \frac{\partial z_{t,2}}{\partial z_{t-1,2}} & \frac{\partial z_{t,2}}{\partial \epsilon_{t,1}} & \frac{\partial z_{t,2}}{\partial \epsilon_{t,2}} \end{bmatrix}.$$

Given that this Jacobian is triangular, we can efficiently compute its determinant as $\prod_i \frac{\partial z_{t,i}}{\epsilon_{t,i}}$. Furthermore, because the noise terms are mutually independent, and hence $\epsilon_{t,i} \perp \epsilon_{t,j}$ for $j \neq i$ and $\epsilon_t \perp \mathbf{z}_{t-1}$, so we can with the RHS of Equation (A25) as follows

$$p(z_{t-1,1}, z_{t-1,2}, z_{t,1}, z_{t,2}) = p(z_{t-1,1}, z_{t-1,2}) \times \frac{p(\epsilon_{t,1}, \epsilon_{t,2})}{|\mathbf{J_f}|} = p(z_{t-1,1}, z_{t-1,2}) \times \frac{\prod_i p(\epsilon_{t,i})}{|\mathbf{J_f}|}. \quad \text{(A26)}$$

Finally, we generalize this example and derive the prior likelihood below. Let $\{r_i\}_{i=1,2,3,\cdots}$ be a set of learned inverse transition functions that take the estimated latent causal variables, and output the noise terms, i.e., $\hat{\epsilon}_{t,i} = r_i(\hat{z}_{t,i}, \{\hat{\mathbf{z}}_{t-\tau}\})$. Then we design a transformation $\mathbf{A} \to \mathbf{B}$ with low-triangular Jacobian as follows:

$$\underbrace{[\hat{\mathbf{z}}_{t-L}, \cdots, \hat{\mathbf{z}}_{t-1}, \hat{\mathbf{z}}_t]^\top}_{\mathbf{A}} \text{ mapped to } \underbrace{[\hat{\mathbf{z}}_{t-L}, \cdots, \hat{\mathbf{z}}_{t-1}, \hat{\epsilon}_{t,i}]^\top}_{\mathbf{B}}, \text{ with } \mathbf{J_{A \to B}} = \begin{bmatrix} \mathbb{I}_{n_s \times L} & 0 \\ * & \text{diag}\left(\frac{\partial r_{i,j}}{\partial \hat{z}_{t,j}}\right) \end{bmatrix}. \quad \text{(A27)}$$

Similar to Equation (A26), we can obtain the joint distribution of the estimated dynamics subspace as:

$$\log p(\mathbf{A}) = \underbrace{\log p(\hat{\mathbf{z}}_{t-L}, \cdots, \hat{\mathbf{z}}_{t-1}) + \sum_{i=1}^{n_s} \log p(\hat{\epsilon}_{t,i}) + \log(|\det(\mathbf{J_{A \to B}})|)}_{\text{Because of mutually independent noise assumption}} \quad \text{(A28)}$$

Finally, we have:

$$\log p(\hat{\mathbf{z}}_t | \{\hat{\mathbf{z}}_{t-\tau}\}_{\tau=1}^L) = \sum_{i=1}^{n_s} p(\hat{\epsilon}_{t,i}) + \sum_{i=1}^{n_s} \log |\frac{\partial r_i}{\partial \hat{z}_{t,i}}| \quad \text{(A29)}$$

Since the prior of $p(\hat{\mathbf{z}}_{t+1:T} | \hat{\mathbf{z}}_{1:t}) = \prod_{i=t+1}^T p(\hat{\mathbf{z}}_i | \hat{\mathbf{z}}_{i-1})$ with the assumption of first-order Markov assumption, we can estimate $p(\hat{\mathbf{z}}_{t+1:T} | \hat{\mathbf{z}}_{1:t})$ in a similar way.

## E.2 EVIDENT LOWER BOUND

In this subsection, we show the evident lower bound. We first factorize the conditional distribution according to the Bayes theorem.

$$\begin{aligned} \ln p(\mathbf{x}_{1:T}) &= \ln \frac{p(\mathbf{x}_{1:T}, \mathbf{z}_{1:T})}{p(\mathbf{z}_{1:T} | \mathbf{x}_{1:T})} = \mathbb{E}_{q(\mathbf{z}_{1:T} | \mathbf{x}_{1:T})} \ln \frac{p(\mathbf{x}_{1:T}, \mathbf{z}_{1:T}) q(\mathbf{z}_{1:T} | \mathbf{x}_{1:T})}{p(\mathbf{z}_{1:T} | \mathbf{x}_{1:T}) q(\mathbf{z}_{1:T} | \mathbf{x}_{1:T})} \\ &\geq \underbrace{\mathbb{E}_{q(\mathbf{z}_{1:T} | \mathbf{x}_{1:T})} \ln p(\mathbf{x}_{1:T} | \mathbf{z}_{1:T})}_{L_r} - \underbrace{D_{KL}(q(\mathbf{z}_{1:T} | \mathbf{x}_{1:T}) || p(\mathbf{z}_{1:T}))}_{L_{KLD}} = ELBO \end{aligned} \quad \text{(A30)}$$

As for the time-series forecasting task, we let $\mathbf{x}_{1:t}$ and $\mathbf{x}_{t+1:T}$ be the historical and future observed variables, then the ELBO can be further derived as follows:

$$\ln p(\mathbf{x}_{t+1:T}, \mathbf{x}_{1:t}) = \ln \frac{p(\mathbf{x}_{t+1:T}, \mathbf{z}_{1:T}, \mathbf{x}_{1:t})}{p(\mathbf{z}_{1:T} | \mathbf{x}_{1:t}, \mathbf{x}_{t+1:T})} = \ln \frac{p(\mathbf{x}_{t+1:T}, \mathbf{z}_{1:t}, \mathbf{z}_{t+1:T}, \mathbf{x}_{1:t})}{p(\mathbf{z}_{1:T} | \mathbf{x}_{1:T})}$$

$$\geq \mathbb{E}_{q(\mathbf{z}_{1:T} | \mathbf{x}_{1:t})} \ln \frac{p(\mathbf{x}_{t+1:T} | \mathbf{z}_{t+1:T}) p(\mathbf{x}_{1:t} | \mathbf{z}_{1:t}) p(\mathbf{z}_{1:T})}{q(\mathbf{z}_{1:T} | \mathbf{x}_{1:t})}$$

$$= \underbrace{\mathbb{E}_{q(\mathbf{z}_{1:T} | \mathbf{x}_{1:t})} \ln p(\mathbf{x}_{1:t} | \mathbf{z}_{1:T})}_{\mathcal{L}_r} + \underbrace{\mathbb{E}_{q(\mathbf{z}_{1:T} | \mathbf{x}_{1:t})} \ln p(\mathbf{x}_{t+1:T} | \mathbf{z}_{1:T})}_{\mathcal{L}_{pre}} - \underbrace{D_{KL}(q(\mathbf{z}_{1:T} | \mathbf{x}_{1:t}) || p(\mathbf{z}_{1:T}))}_{\mathcal{L}_{KLD}},$$

$$\text{(A31)}$$

where $\mathcal{L}_r$ denotes the reconstruct loss of historical data, $\mathcal{L}_{pre}$ denotes the forecasting loss of future data.

## E.3 MODEL DETAILS

We choose MICN Wang et al. (2022) as the encoder backbone of our model on real-world datasets. Specifically, given the MICN extracts the hidden feature, we apply a variational inference block and then an MLP-based decoder. Architecture details of the proposed method are shown in Table A4.

Table A4: Architecture details. $T$, length of time series. $|\mathbf{x}_t|$: input dimension. $n$: latent dimension. LeakyReLU: Leaky Rectified Linear Unit. Tanh: Hyperbolic tangent function.

| Configuration | Description | Output |
|---|---|---|
| $\phi$ | Latent Variable Encoder | |
| Input:$\mathbf{x}_{1:t}$ | Observed time series | Batch Size$\times$t$\times$ $\mathbf{x}$ dimension |
| Dense | $|\mathbf{x}_t|$ neurons | Batch Size$\times$t$\times|\mathbf{x}_t|$ |
| Concat zero | concatenation | Batch Size$\times$T$\times|\mathbf{x}_t|$ |
| Dense | n neurons | Batch Size$\times$T$\times$n |
| $\psi$ | Decoder | |
| Input:$\mathbf{z}_{1:T}$ | Latent Variable | Batch Size$\times$T$\times$n |
| Dense | $|\mathbf{x}_t|$ neurons, Tanh | Batch Size$\times$T$\times|\mathbf{x}_t|$ |
| r | Modular Prior Networks | |
| Input: $\mathbf{z}_{1:T}$ | Latent Variable | Batch Size$\times$(n+1) |
| Dense | 128 neurons,LeakyReLU | (n+1)$\times$128 |
| Dense | 128 neurons,LeakyReLU | 128$\times$128 |
| Dense | 128 neurons,LeakyReLU | 128$\times$128 |
| Dense | 1 neuron | Batch Size$\times$1 |
| Jacobian Compute | Compute log(det(J)) | Batch Size |

### E.3.1 REPRODUCIBILITY OF SIMULATION EXPERIMENTS.

For the implementation of baseline models, we utilized publicly released code for TDRL and iCRITIS. However, since the author did not release the code for G-CaRL, we implemented it ourselves based on the paper. It is important to note that the original code of iCRITIS only accepts images as input. To adapt it to our needs, we replaced the encoder and decoder with a Variational Autoencoder, with the same hyperparameters used in IDOL. Our code is modified based on the code of TDRL, and shared hyperparameters remain the same.

When calculating the MCC for all methods, we use the mean value from the VAE encoder. Regarding the latent causal graph, we employ different approaches depending on the method. For IDOL, TDRL, and G-CaRL, we utilize the Jacobian matrix as a proxy for capturing the causal process. On the other hand, for iCRITIS, we rely on the "get_adj_matrix()" function provided in its original code to obtain the latent causal graph.

### E.3.2 REPRODUCIBILITY OF REAL-WORLD EXPERIMENTS.

The model details of our method are shown in Table A4. For a fair comparison, we employ the official implementations and use the default hyperparameters. Since our model is modified from the official implementations of TDRL, both our model and the baselines share the same hyperparameters. To achieve the results from the well-fit baselines, we employed the default hyper-parameters and tried different values of learning rate for the best models. Please refer to Table A5 to A11 for the implementation details of the baselines.

Table A5: TDRL architecture details. $T$: length of time series. $|x_t|$: input dimension. $n$: latent dimension. LeakyReLU: Leaky Rectified Linear Unit. Tanh: Hyperbolic tangent function.

| Configuration | Description | Output |
|---|---|---|
| $\phi$ | Latent Variable Encoder | |
| Input:$x_{1:t}$ | Observed time series | Batch Size×t× X dimension |
| Dense | $|x_t|$ neurons | Batch Size×t×$|x_t|$ |
| Concat zero | concatenation | Batch Size×T×$|x_t|$ |
| Dense | n neurons | Batch Size×T×n |
| $\psi$ | Decoder | |
| Input:z1:T | Latent Variable | Batch Size×T×n |
| Dense | $|x_t|$ neurons,Tanh | Batch Size×T×$|x_t|$ |
| r | Modular Prior Networks | |
| Input:z1:T | Latent Variable | Batch Size×(n+1) |
| Dense | 128 neurons,LeakyReLU | (n+1)×128 |
| Dense | 128 neurons,LeakyReLU | 128×128 |
| Dense | 128 neurons,LeakyReLU | 128×128 |
| Dense | 1 neuron | Batch Size×1 |
| JacobianCompute | Compute log(det(J)) | Batch Size |

Table A6: CARD architecture details. $T$: length of time series. $|x_t|$: input dimension. $p$: patch number. $n$: patch length.

| Configuration | Description | Output |
|---|---|---|
| Input:$x_{1:t}$ | Observed time series | Batch Size×t× X dimension |
| Permute | Matrix Transpose | BS×$|x_t|$ × t |
| unfold | unfold | BS×$|x_t|$ × p×n |
| Add random | add | BS×$|x_t|$×p×n |
| Dense | d neurons, dropout | BS×$|x_t|$×p×d |
| Concat random | concatenation | BS×$|x_t|$×(p+1)×d |
| Dense | Attenion | BS×$|x_t|$×(p+1)×d |
| reshape | reshape | BS×$|x_t|$×((p+1)×d) |
| Dense | (T-t) neurons | BS×$|x_t|$×(T-t) |
| Permute | Matrix Transpose | BS×(T-t)×$|x_t|$ |

Table A7: iTransformer architecture details. $T$: length of time series. $|x_t|$: input dimension.

| Configuration | Description | Output |
|---|---|---|
| Input:$x_{1:t}$ | Observed time series | Batch Size×t× X dimension |
| Permute | Matrix Transpose | BS×$|x_t|$×t |
| Dense | Embedding | BS×$|x_t|$×d |
| Dense | Attention | BS×$|x_t|$×d |
| Dense | T-t neurons | BS×$|x_t|$×(T-t) |
| Permute | Matrix Transpose | BS×(T-t)×$|x_t|$ |

Table A8: Auoformer architecture details. $T$: length of time series. $|x_t|$: input dimension.

| Configuration | Description | Output |
|---|---|---|
| Input:$x_{1:t}$ | Observed time series | Batch Size×t× X dimension |
| seasonal, trend | AvgPool1d | BS×t×$|x_t|$ |
| seasonal concat zero | concatenation | BS×T×$|x_t|$ |
| trend concat x_mean | concatenation | BS×T×$|x_t|$ |
| seasonal | Embedding | BS×T×d |
| Input:$x_{1:t}$ | Observed time series | Batch Size×t× X dimension |
| Dense | Embedding | BS×t×d |
| x_enc | Attention | BS×t×d |
| Input:x_enc, seasonal | Conv1d | BS×T×d |
| Dense | Conv1d | BS×T×$|x_t|$ |
| Add trend | Add | BS×T×$|x_t|$ |

Table A9: TimesNet architecture details. $T$: length of time series. $|x_t|$: input dimension.

| Configuration | Description | Output |
|---|---|---|
| Input:$x_{1:t}$ | Observed time series | Batch Size×t× X dimension |
| Dense | Embedding | BS×t×d |
| Permute | Matrix Transpose | BS×d×t |
| Dense | T neurons | BS×d×T |
| Permute | Matrix Transpose | BS×T×d |
| Dense | Conv1d,LayerNorm | BS×T×d |
| Dense | $|x_t|$ neurons | BS×T×$|x_t|$ |

Table A10: MICN architecture details. $T$: length of time series. $|x_t|$: input dimension. LeakyReLU: Leaky Rectified Linear Unit. Tanh: Hyperbolic tangent function.

| Configuration | Description | Output |
|---|---|---|
| Input:$x_{1:t}$ | Observed time series | Batch Size×t× X dimension |
| seasonal, trend | AvgPool1d | BS×t×$|x_t|$ |
| seasonal concat zero | concatenation | BS×T×$|x_t|$ |
| trend permute | Matrix Transpose | BS×$|x_t|$×t |
| trend | (T-t) neurons | BS×$|x_t|$×(T-t) |
| trend permute | Matrix Transpose | BS×(T-t)×$|x_t|$ |
| seasonal | d neurons | BS×T×d |
| seasonal | Conv1d,LayerNorm,Tanh | BS×T×d |
| seasonal | $|x_t|$ neurons | BS×T×$|x_t|$ |
| seasonal add trend | Add | BS×(T-t)×$|x_t|$ |

Table A11: FITS architecture details. $T$: length of time series. $|x_t|$: input dimension. $p$: patch number. $n$: patch length.

| Configuration | Description | Output |
|---|---|---|
| Input:$x_{1:t}$ | Observed time series | Batch Size×t× X dimension |
| FFT | rfft | BS×(t/2+1)×$|x_t|$ |
| Permute | Matrix Transpose | BS×$|x_t|$×(t/2+1) |
| Dense | T/2+1 neurons | BS×$|x_t|$×(T/2+1) |
| Permute | Matrix Transpose | BS×(T/2+1)×$|x_t|$ |
| IFFT | irfft | BS×T×$|x_t|$ |

Table A12: Standard Deviation results on simulation data.

| Datasets | IDOL | TDRL | G-CaRL | iCITRIS | $\beta-$VAE | SlowVAE | iVAE | FactorVAE | PCL | TCL |
|---|---|---|---|---|---|---|---|---|---|---|
| A | 0.029 | 0.045 | 0.003 | 0.105 | 0.014 | 0.025 | 0.064 | 0.025 | 0.011 | 0.008 |
| B | 0.013 | 0.022 | 0.004 | 0.039 | 0.013 | 0.005 | 0.011 | 0.035 | 0.073 | 0.028 |
| C | 0.002 | 0.028 | 0.003 | 0.039 | 0.031 | 0.010 | 0.024 | 0.027 | 0.021 | 0.038 |
| D | 0.006 | 0.003 | 0.051 | 0.010 | 0.028 | 0.074 | 0.034 | 0.027 | 0.033 | 0.007 |
| E | 0.037 | 0.005 | 0.001 | 0.021 | 0.015 | 0.002 | 0.009 | 0.002 | 0.039 | 0.016 |
| F | 0.029 | 0.018 | 0.004 | - | 0.165 | 0.134 | 0.014 | 0.031 | 0.018 | 0.007 |

Table A13: MCC results of IDOL and IDOL-S with mean and standard deviation.

| | IDOL | IDOL-S |
|---|---|---|
| MCC | 0.8595 (0.0599) | 0.8498 (0.0481) |

# F EXPERIMENT DETAILS

## F.1 SIMULATION EXPERIMENT

### F.1.1 DATA GENERATION PROCESS

As for the temporally latent processes, we use MLPs with the activation function of LeakyReLU to model the sparse time-delayed and instantaneous relationships of temporally latent variables. For all datasets, we set sequence length as 5 and transition lag as 1. That is:

$$z_{t,i} = (LeakyReLU(W_{i,:} \cdot \mathbf{z}_{t-1}, 0.2) + V_{<i,i} \cdot \mathbf{z}_{t,<i}) \cdot \epsilon_{t,i} + \epsilon_{t,i},$$

where $W_{i,:}$ is the $i$-th row of $W$ and $V_{<i,i}$ is the first $i-1$ columns in the $i$-th row of $V$. Moreover, each independent noise $\epsilon_{t,i}$ is sampled from the distribution of normal distribution. We further let the data generation process from latent variables to observed variables be MLPs with the LeakyReLU units.

We provide 6 synthetic datasets, with 3,5,8,8,8,16 latent variables from A to F. For dataset A, we have $W_A = [[1, 1, 0], [0, 1, 0], [0, 0, 1]]$. For dataset $B$, we have $W_B$ as an eye matrix with 2 extra nonzero entries. For dataset $C, F$, we have $W_C, W_F$ as eye matrices. For datasets $D$ and $E$, $W_D$ and $W_E$ are dense matrices with all nonzero entries, which are generated by the data generator of TDRL(Yao et al., 2022). When it comes to $V$, we have and only have $V_{i-1,i} = 1 \ \forall i > 0$ for dataset $A, B, C, E, F$. We also set $V_D = \mathbf{0}$, which means that there are no instantaneous effects.

The total size of the dataset is 100,000, with 1,024 samples designated as the validation set. The remaining samples are the training set.

### F.1.2 EVALUATION METRICS.

To evaluate the identifiability performance of our method under instantaneous dependencies, we employ the Mean Correlation Coefficient (MCC) between the ground-truth $\mathbf{z}_t$ and the estimated $\hat{\mathbf{z}}_t$. A higher MCC denotes a better identification performance the model can achieve. In addition, we also draw the estimated latent causal process to validate our method. Since the estimated transition function will be a transformation of the ground truth, we do not compare their exact values, but only the activated entries.

### F.1.3 MORE SIMULATION EXPERIMENT RESULTS

We run each experiment 3 times, with seeds $769, 770, 771$. Standard deviation results of the simulation datasets are shown in Table A12.

### F.1.4 ABLATION STUDY

To further illustrate the significance of sparsity, we compare our IDOL model with its variant that lacks a sparsity constraint (IDOL-S). The experimental results on the synthetic dataset are presented in Table A13. The result shows that sparsity constraint is crucial to the model.

Table A14: MSE and MAE results of different methods on the transformed HumanEva-I datase.

| dataset | Predict Length | IDOL | | TDRL | | CARD | | FITS | | MICN | | iTransformer | | TimesNet | | Autoformer | |
|---|---|---|---|---|---|---|---|---|---|---|---|---|---|---|---|---|---|
| | | MSE | MAE | MSE | MAE | MSE | MAE | MSE | MAE | MSE | MAE | MSE | MAE | MSE | MAE | MSE | MAE |
| D | 125 | **0.082** | **0.196** | 0.084 | 0.202 | 0.127 | 0.251 | 0.108 | 0.237 | 0.083 | 0.201 | 0.106 | 0.227 | 0.111 | 0.235 | 0.133 | 0.267 |
| | 250 | **0.101** | **0.226** | 0.134 | 0.274 | 0.204 | 0.319 | 0.141 | 0.273 | 0.107 | 0.236 | 0.161 | 0.287 | 0.192 | 0.312 | 0.188 | 0.326 |
| | 375 | **0.113** | **0.244** | 0.144 | 0.288 | 0.230 | 0.342 | 0.180 | 0.306 | 0.115 | 0.247 | 0.193 | 0.315 | 0.226 | 0.344 | 0.224 | 0.352 |
| G | 125 | **0.091** | **0.220** | 0.097 | 0.235 | 0.098 | 0.227 | 0.168 | 0.305 | 0.095 | 0.225 | 0.097 | 0.223 | 0.117 | 0.247 | 0.141 | 0.283 |
| | 250 | **0.103** | **0.254** | 0.139 | 0.285 | 0.144 | 0.274 | 0.183 | 0.320 | 0.130 | 0.272 | 0.123 | 0.260 | 0.134 | 0.271 | 0.154 | 0.300 |
| | 375 | **0.138** | **0.276** | 0.155 | 0.296 | 0.179 | 0.310 | 0.182 | 0.321 | 0.154 | 0.295 | 0.146 | 0.286 | 0.165 | 0.308 | 0.161 | 0.308 |
| P | 125 | **1.195** | **0.680** | 1.337 | 0.834 | 1.531 | 0.787 | 2.306 | 1.027 | 1.259 | 0.786 | 1.488 | 0.793 | 1.905 | 0.911 | 2.941 | 1.210 |
| | 250 | **1.961** | **1.045** | 2.411 | 1.307 | 3.124 | 1.346 | 3.221 | 1.293 | 2.498 | 1.346 | 3.152 | 1.383 | 3.908 | 1.513 | 4.169 | 1.633 |
| | 375 | **2.326** | **1.164** | 2.514 | 1.326 | 3.948 | 1.558 | 4.020 | 1.537 | 2.377 | 1.274 | 4.075 | 1.631 | 4.315 | 1.662 | 4.951 | 1.815 |
| SD | 125 | **0.365** | **0.405** | 0.373 | 0.418 | 0.512 | 0.482 | 0.595 | 0.543 | 0.379 | 0.425 | 0.517 | 0.490 | 0.523 | 0.501 | 0.785 | 0.632 |
| | 250 | **0.551** | **0.526** | 0.573 | 0.540 | 0.874 | 0.671 | 0.923 | 0.710 | 0.569 | 0.532 | 0.813 | 0.655 | 0.815 | 0.656 | 1.061 | 0.769 |
| | 375 | **0.649** | **0.579** | 0.653 | 0.581 | 0.980 | 0.720 | 1.075 | 0.781 | 0.655 | 0.587 | 0.959 | 0.721 | 0.966 | 0.726 | 1.193 | 0.826 |
| W | 125 | **0.045** | **0.162** | 0.047 | 0.164 | 0.122 | 0.254 | 0.288 | 0.431 | 0.046 | **0.162** | 0.124 | 0.257 | 0.067 | 0.184 | 0.314 | 0.405 |
| | 250 | **0.104** | **0.256** | 0.143 | 0.302 | 0.333 | 0.434 | 0.589 | 0.618 | 0.141 | 0.302 | 0.340 | 0.438 | 0.170 | 0.292 | 0.629 | 0.614 |
| | 375 | **0.150** | **0.313** | 0.167 | 0.331 | 0.431 | 0.508 | 0.691 | 0.673 | 0.175 | 0.341 | 0.425 | 0.506 | 0.253 | 0.370 | 1.048 | 0.761 |
| WT | 125 | **0.062** | **0.189** | 0.078 | 0.214 | 0.118 | 0.262 | 0.227 | 0.385 | 0.071 | 0.204 | 0.115 | 0.256 | 0.131 | 0.271 | 0.223 | 0.361 |
| | 250 | **0.127** | **0.277** | 0.152 | 0.311 | 0.297 | 0.421 | 0.407 | 0.520 | 0.153 | 0.312 | 0.287 | 0.414 | 0.195 | 0.338 | 0.395 | 0.509 |
| | 375 | **0.151** | **0.305** | 0.171 | 0.328 | 0.365 | 0.473 | 0.425 | 0.530 | 0.169 | 0.327 | 0.313 | 0.441 | 0.289 | 0.431 | 0.430 | 0.538 |

Table A15: Standard deviation of MSE and MAE results on the different motion.

| dataset | | Predict Length | IDOL | | TDRL | | CARD | | FITS | | MICN | | iTransformer | | TimesNet | | Autoformer | |
|---|---|---|---|---|---|---|---|---|---|---|---|---|---|---|---|---|---|---|
| | | | MSE | MAE | MSE | MAE | MSE | MAE | MSE | MAE | MSE | MAE | MSE | MAE | MSE | MAE | MSE | MAE |
| H36M | G | 100 | 0.0008 | 0.0004 | 0.0018 | 0.0026 | 0.0008 | 0.0010 | 0.0008 | 0.0010 | 0.0007 | 0.0027 | 0.0016 | 0.0019 | 0.0038 | 0.0049 | 0.0008 | 0.0015 |
| | | 125 | 0.0009 | 0.0012 | 0.0009 | 0.0010 | 0.0006 | 0.0008 | 0.0005 | 0.0008 | 0.0017 | 0.0033 | 0.0008 | 0.0009 | 0.0004 | 0.0015 | 0.0012 | 0.0012 |
| | | 150 | 0.0021 | 0.0022 | 0.0026 | 0.0029 | 0.0004 | 0.0005 | 0.0002 | 0.0003 | 0.0002 | 0.0008 | 0.0001 | 0.0003 | 0.0033 | 0.0023 | 0.0012 | 0.0011 |
| | J | 125 | 0.0144 | 0.0217 | 0.0115 | 0.0085 | 0.0499 | 0.0263 | 0.0026 | 0.0004 | 0.0032 | 0.0081 | 0.0256 | 0.0131 | 0.0872 | 0.0296 | 0.0238 | 0.0171 |
| | | 150 | 0.0099 | 0.0161 | 0.0088 | 0.0083 | 0.0108 | 0.0050 | 0.0023 | 0.0010 | 0.0106 | 0.0135 | 0.0183 | 0.0116 | 0.0837 | 0.0325 | 0.0085 | 0.0089 |
| | | 175 | 0.0003 | 0.0038 | 0.0126 | 0.0123 | 0.0478 | 0.0130 | 0.0030 | 0.0010 | 0.0088 | 0.0129 | 0.0355 | 0.0200 | 0.0289 | 0.0109 | 0.0603 | 0.0133 |
| | TC | 25 | 0.0001 | 0.0005 | 0.0009 | 0.0028 | 0.0009 | 0.0034 | 0.0009 | 0.0022 | 0.0001 | 0.0010 | 0.0001 | 0.0004 | 0.0015 | 0.0023 | 0.0026 | 0.0055 |
| | | 50 | 0.0004 | 0.0022 | 0.0011 | 0.0023 | 0.0018 | 0.0038 | 0.0003 | 0.0010 | 0.0003 | 0.0013 | 0.0001 | 0.0004 | 0.0017 | 0.0046 | 0.0021 | 0.0042 |
| | | 75 | 0.0005 | 0.0026 | 0.0007 | 0.0022 | 0.0019 | 0.0030 | 0.0001 | 0.0002 | 0.0004 | 0.0015 | 0.0002 | 0.0004 | 0.0029 | 0.0078 | 0.0031 | 0.0066 |
| | W | 25 | 0.0019 | 0.0039 | 0.0016 | 0.0020 | 0.0053 | 0.0046 | 0.0061 | 0.0069 | 0.0005 | 0.0012 | 0.0018 | 0.0015 | 0.0023 | 0.0027 | 0.0887 | 0.0796 |
| | | 50 | 0.0003 | 0.0009 | 0.0020 | 0.0019 | 0.0432 | 0.0136 | 0.0009 | 0.0004 | 0.0005 | 0.0005 | 0.0022 | 0.0023 | 0.0064 | 0.0078 | 0.0660 | 0.0419 |
| | | 75 | 0.0154 | 0.0217 | 0.0011 | 0.0019 | 0.0630 | 0.0131 | 0.0033 | 0.0025 | 0.0009 | 0.0014 | 0.0033 | 0.0017 | 0.0320 | 0.0169 | 0.0499 | 0.0349 |
| HumanEVA-I | D | 125 | 0.0002 | 0.0028 | 0.0011 | 0.0015 | 0.0012 | 0.0020 | 0.0013 | 0.0013 | 0.0010 | 0.0017 | 0.0015 | 0.0015 | 0.0048 | 0.0037 | 0.0007 | 0.0008 |
| | | 250 | 0.0012 | 0.0016 | 0.0024 | 0.0024 | 0.0067 | 0.0038 | 0.0002 | 0.0005 | 0.0010 | 0.0013 | 0.0009 | 0.0006 | 0.0128 | 0.0089 | 0.0044 | 0.0042 |
| | | 375 | 0.0056 | 0.0062 | 0.0001 | 0.0009 | 0.0109 | 0.0064 | 0.0003 | 0.0004 | 0.0020 | 0.0026 | 0.0033 | 0.0021 | 0.0063 | 0.0012 | 0.0043 | 0.0030 |
| | G | 125 | 0.0043 | 0.0090 | 0.0066 | 0.0111 | 0.0026 | 0.0030 | 0.0058 | 0.0050 | 0.0064 | 0.0105 | 0.0053 | 0.0054 | 0.0016 | 0.0019 | 0.0076 | 0.0066 |
| | | 250 | 0.0095 | 0.0174 | 0.0105 | 0.0147 | 0.0093 | 0.0059 | 0.0027 | 0.0022 | 0.0016 | 0.0021 | 0.0063 | 0.0071 | 0.0006 | 0.0018 | 0.0003 | 0.0002 |
| | | 375 | 0.0014 | 0.0017 | 0.0005 | 0.0006 | 0.0059 | 0.0034 | 0.0013 | 0.0004 | 0.0037 | 0.0047 | 0.0085 | 0.0079 | 0.0035 | 0.0036 | 0.0046 | 0.0043 |
| | P | 125 | 0.0311 | 0.0277 | 0.0951 | 0.0434 | 0.0465 | 0.0229 | 0.0169 | 0.0062 | 0.0333 | 0.0277 | 0.0251 | 0.0032 | 0.2698 | 0.0744 | 0.0700 | 0.0013 |
| | | 250 | 0.0719 | 0.0225 | 0.0601 | 0.0318 | 0.1735 | 0.0154 | 0.0317 | 0.0086 | 0.0155 | 0.0069 | 0.0123 | 0.0024 | 0.5525 | 0.0976 | 0.0708 | 0.0093 |
| | | 375 | 0.2728 | 0.0168 | 0.0889 | 0.0335 | 0.1108 | 0.0319 | 0.0051 | 0.0907 | 0.0382 | 0.0171 | 0.0222 | 0.0034 | 0.0472 | 0.0040 | 0.0880 | 0.0243 |
| | SD | 125 | 0.0056 | 0.0031 | 0.0119 | 0.0116 | 0.0045 | 0.0038 | 0.0043 | 0.0027 | 0.0157 | 0.0155 | 0.0131 | 0.0099 | 0.0219 | 0.0167 | 0.0362 | 0.0235 |
| | | 250 | 0.0024 | 0.0002 | 0.0075 | 0.0077 | 0.0415 | 0.0176 | 0.0008 | 0.0011 | 0.0229 | 0.0146 | 0.0064 | 0.0029 | 0.0172 | 0.0069 | 0.0061 | 0.0017 |
| | | 375 | 0.0149 | 0.0145 | 0.0072 | 0.0079 | 0.0107 | 0.0044 | 0.0025 | 0.0014 | 0.0050 | 0.0020 | 0.0110 | 0.0057 | 0.0105 | 0.0051 | 0.0336 | 0.0100 |
| | W | 125 | 0.0029 | 0.0043 | 0.0078 | 0.0150 | 0.0034 | 0.0041 | 0.0037 | 0.0039 | 0.0016 | 0.0034 | 0.0062 | 0.0087 | 0.0028 | 0.0032 | 0.1585 | 0.1221 |
| | | 250 | 0.0082 | 0.0105 | 0.0236 | 0.0300 | 0.0157 | 0.0127 | 0.0002 | 0.0006 | 0.0125 | 0.0142 | 0.0130 | 0.0101 | 0.0105 | 0.0107 | 0.1344 | 0.0748 |
| | | 375 | 0.0089 | 0.0108 | 0.0273 | 0.0296 | 0.0035 | 0.0053 | 0.0019 | 0.0014 | 0.0212 | 0.0225 | 0.0121 | 0.0088 | 0.0081 | 0.0067 | 0.2889 | 0.0771 |
| | WT | 125 | 0.0031 | 0.0029 | 0.0065 | 0.0106 | 0.0059 | 0.0066 | 0.0032 | 0.0035 | 0.0019 | 0.0022 | 0.0021 | 0.0027 | 0.0056 | 0.0042 | 0.0453 | 0.0455 |
| | | 250 | 0.0014 | 0.0030 | 0.0044 | 0.0066 | 0.0123 | 0.0102 | 0.0007 | 0.0010 | 0.0054 | 0.0085 | 0.0113 | 0.0111 | 0.0554 | 0.0523 | 0.0168 | 0.0152 |
| | | 375 | 0.0146 | 0.0188 | 0.0020 | 0.0027 | 0.0431 | 0.0283 | 0.0019 | 0.0014 | 0.0029 | 0.0027 | 0.0077 | 0.0054 | 0.0328 | 0.0246 | 0.0054 | 0.0043 |

Table A16: Sample number of different datasets.

| Dataset | | Size |
|---|---|---|
| Humaneva | Gestures | 1686 |
| | Jog | 2050 |
| | ThrowCatch | 1021 |
| | Walking | 2454 |
| Human | Discussion | 13759 |
| | Greeting | 8416 |
| | Purchases | 7480 |
| | SittingDown | 16438 |
| | Walking | 16257 |
| | WalkTogether | 10658 |

## F.2   REAL-WORLD EXPERIMENTS

### F.2.1   REAL-WORLD DATASET DESCRIPTION

**Human3.6M Dataset** is collected over 3.6 million different human poses, viewed from 4 different angles, using an accurate human motion capture system. The motions were executed by 11 professional actors, and cover a diverse set of everyday scenarios including conversations, eating, greeting, talking on the phone, posing, sitting, smoking, taking photos, waiting, walking in various non-typical scenarios. We randomly use four motions, i.e., Gestures (Ge), Jog (J), CatchThrow (CT), and Walking (W), for the task of the human motion forecasting. The data is obtained from the joint angles (provided by Vicon's skeleton fitting procedure) by applying forward kinematics on the skeleton of the subject. The parametrization is called relative because there is a specially designated joint, usually called the root (roughly corresponding to the pelvis bone position), which is taken as the center of the coordinate system, while the other joints are estimated relative to it.

In this dataset, the joint positions can be considered as latent variables. And the kinematic representation can be considered as the observed variables. The kinematic representation (KR) considers the relative joint angles between limbs. We consider the process from joint position to joint angles as a mixture process.

**HumanEva-I dataset** comprises 3 subjects each performing several action categories. Each pose has 15 joints with three axis. We choose 6 motions, i.e., : Discussion (D), Greeting (Gr), Purchases (P), SittingDown (SD), Walking (W), and WalkTogether (WT) for the task of human motion forecasting. Specifically, the ground truth motion of the body was captured using a commercial motion capture (MoCap) system from ViconPeak.5 The system uses reflective markers and six 1M-pixel cameras to recover the 3D position of the markers and thereby estimate the 3D articulated pose of the body. We consider the joints as latent variables and the signals recorded from the system as observations.

The sample number of different datasets are shown in Table A16.

### F.2.2   MORE EXPERIMENTS RESULTS

We further consider a more complex mixture process. To achieve is, we further apply a transformation on the observed variables, i.e., $\bar{\mathbf{x}}_t = f_o(\mathbf{x}_t)$, where $f_o$ is a linear transformation. Then we can consider the sensors as latent variables with instantaneous dependencies and conduct motion forecasting on the transformed datasets. Experiment results on the transformed HumanEva-I dataset are shown in Table A14.

### F.2.3   QUALITATIVE RESULTS

To further show the effectiveness of the proposed method, we also randomly select a batch of test samples and visualize the forecasting results of different baselines. Visualization results in Figure A9 show the predicted results of "Discussion", "Greeting", and "Purchases", respectively, which illustrate how the forecasting results align the ground-truth motions, where the red lines denote the ground-truth motions and the lines in other colors denote the forecasting results of different methods. According to the experiment results, we can find that the forecasting results of our method achieve

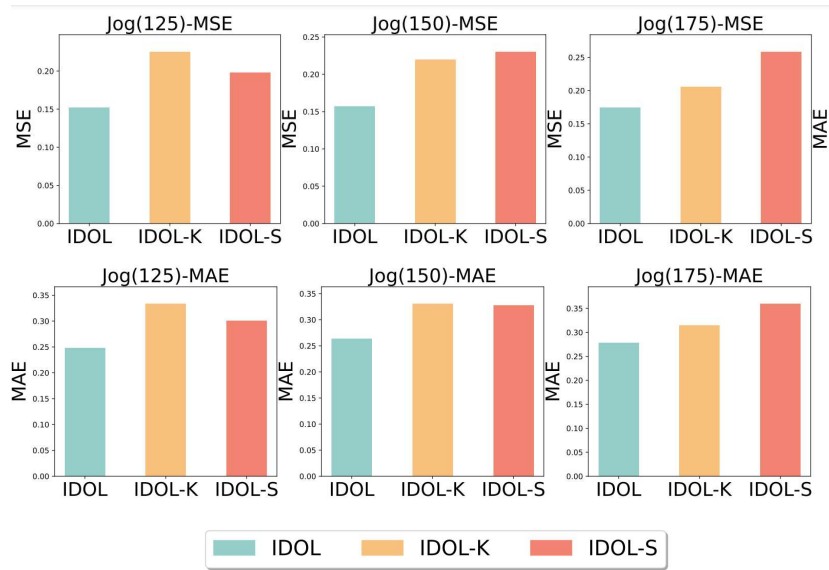

Figure A8: Ablation study on the Jog motion. we explore the impact of different loss terms.

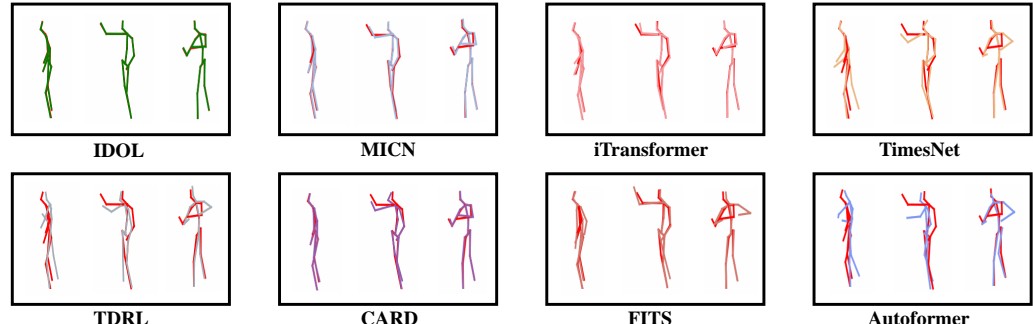

Figure A9: The illustration of visualization of different methods of Walking. The red lines denote the ground-truth motions, those lines in other colors denote the prediction motions of different methods.

the best alignment of the ground truth. In the meanwhile, the other methods that do not consider the instantaneous dependencies of latent variables may generate some exaggerated body movements, for example, the hyperflexed knee joint and the actions that violate the physics rules, which reflects the importance of considering the instantaneous dependencies of latent causal process of time series data.

#### F.2.4 ABLATION STUDY

To evaluate the effectiveness of individual loss terms, we also devise the two model variants. 1) **IDOL-K**: remove the KL divergence restriction of prior estimation. 2) **IDOL-S**: remove sparsity restriction of latent dynamics. Experiment results on the Jog dataset are shown in Figure A8. We can find that the sparsity restriction of latent dynamics plays an important role in the model performance, reflecting that the restriction of latent dynamics can benefit the identifiability of latent variables. We also discover that incorporating the KL divergence has a positive impact on the overall performance of the model, which shows the necessity of identifiability.

## G COMPLEXITY ANALYSIS

### G.1 WALL-CLOCK TRAINING TIMES

To evaluate the computational complexity of our method, we provide the wall-clock training times of different methods. Specifically, we use a consistent hardware setup including the same GPU, CPU, and memory configurations for each model to ensure comparability. Sequentially, in our codes, we

Table A17: Mean and standard deviation of wall-clock training times for one epoch.

| Methods | IDOL | TDRL | CARD | FITS | MICN | iTransformer | TimesNet | Autoformer |
|---------|------|------|------|------|------|--------------|----------|------------|
| Second | 65.960(7.556) | 33.902(5.781) | 62.994(3.035) | 92.941(0.715) | 45.547(13.034) | 38.155(6.441) | 324.941(25.286) | 51.648(15.673) |

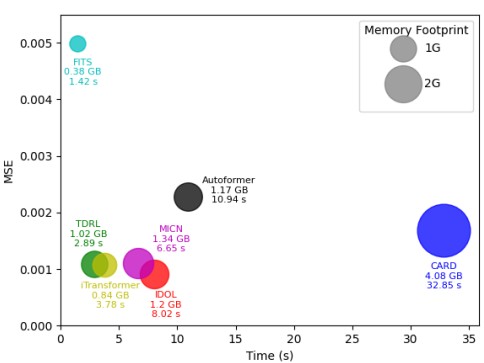

(a) Model Efficiency on the Low-dimension Dataset (Human Dataset).

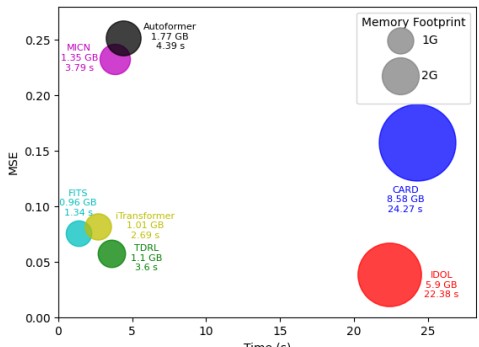

(b) Model Efficiency on the High-dimension Dataset (MoCap Dataset).

Figure A10: Model efficiency comparison for one training step on different datasets.

wrap the training process in a timer to measure the actual elapsed time from start to finish of one epoch. For each method, we repeat three different times and further report the mean and standard deviation. The wall-clock training times are shown in Table A17.

Based on the experimental results, we draw the following conclusions:

- Compared to our baseline model TDRL, the wall-clock training time of the proposed IDOL is nearly twice as slow. Theoretically speaking, the primary computational difference is that IDOL calculates the Jacobian matrix for both time-delayed and instantaneous relationships, while TDRL only calculates the time-delayed component.
- Compared to other mainstream baselines, our IDOL method is slower than some models like MICN and iTransformer. However, our method is still faster than models like FITS.

## G.2 MODEL EFFICIENCY ANALYSIS

To evaluate the model efficiency, we provide model efficiency comparison on the low-dimension dataset (e.g., the Human dataset) and the high-dimension dataset (the MoCap dataset) by evaluating the model efficiency from three aspects: forecasting performance, training speed, and memory footprint. As shown in Figure A10, in low-dimensional datasets, IDOL performs nearly as efficiently as top methods like Autoformer and MICN and outperforms others like CARD. However, in high-dimensional datasets (117 observations), IDOL requires more training time due to the added complexity of Jacobian calculations for instantaneous effects.

## H MORE DISCUSSIONS

### H.1 ANALYSIS OF HIGH-DIMENSIONAL DATA

Identifying causal relationships in high-dimensional latent variable settings is a well-recognized challenge in the causal inference community (Lopez et al., 2022; Cheng et al., 2024). As the dimensionality increases, the complexity of identifying causal structures grows due to the expanding search space. We conduct experiments on both synthetic and real-world datasets to verify it and propose a potential solution.

### H.1.1 SIMULATION EXPERIMENTS

As for the experiment on the synthetic datasets, we follow the same data generation process to generate simulation datasets with latent variable dimensions of 8, 16, 24, and 32. All these datasets

Table A18: Experiments results of IDOL on simulation data on different dimensions of latent variables.

| Dimension | 8 | 16 | 24 | 32 |
|---|---|---|---|---|
| MCC | 0.9801(0.002) | 0.9747 (0.0029) | 0.9243 (0.0173) | 0.8640 (0.0071) |

Table A19: Experiment results of the CMU MoCap datasets.

| Motion | Predicted Length | IDOL | | TDRL | | CARD | | FITS | | MICN | | iTransformer | | TimesNet | | Autoformer | |
|---|---|---|---|---|---|---|---|---|---|---|---|---|---|---|---|---|---|
| | | MSE | MAE | MSE | MAE | MSE | MAE | MSE | MAE | MSE | MAE | MSE | MAE | MSE | MAE | MSE | MAE |
| Running | 50-25 | 0.110 | 0.082 | 0.448 | 0.108 | 0.998 | 0.135 | **0.076** | **0.064** | 0.658 | 0.135 | 0.458 | 0.106 | 2.616 | 0.202 | 1.033 | 0.179 |
| | 50-50 | 0.286 | 0.109 | 1.779 | 0.167 | 2.217 | 0.153 | **0.266** | **0.103** | 0.978 | 0.161 | 1.831 | 0.170 | 4.720 | 0.252 | 3.944 | 0.283 |
| Soccer | 50-25 | **0.022** | **0.043** | 0.026 | 0.047 | 0.206 | 0.082 | 0.076 | 0.065 | 0.057 | 0.066 | 0.032 | 0.051 | 0.063 | 0.068 | 0.211 | 0.105 |
| | 50-50 | **0.079** | **0.071** | 0.084 | 0.073 | 0.397 | 0.108 | 0.265 | 0.103 | 0.284 | 0.120 | 0.133 | 0.082 | 0.392 | 0.107 | 0.452 | 0.143 |

share a similar latent causal process: chain-like instantaneous effects and ono-to-one temporal effects. We then measured the Mean Correlation Coefficient (MCC) between the ground truth $\mathbf{z}_t$ and the estimated $\hat{\mathbf{z}}_t$. The experimental results are presented in Table A18. According to the experiment result, as the dimension of latent variables increases, the value of MCC is still acceptable, despite possible performance loss in high-dimensional problems.

### H.1.2 REAL-WORLD EXPERIMENTS

For the real-world datasets, we consider the CMU-MoCap dataset[3]. It contains various motion capture recordings and 117 skeleton-based measurements, which is a higher dimensionality compared to the Human dataset. We choose the Running and Soccer motions and take the input-prediction length as 50-25 and 50-50, respectively. As shown in Table A19, the IDOL model achieved a comparable forecasting performance in the high-dimensional dataset Running.

### H.1.3 POTENTIAL SOLUTION FOR HIGH-DIMENSION DATA

To better address this challenge, here we propose several potential solutions to more effectively address the challenges of high-dimensional time-series data. One idea is to make use of the divide-and-conquer strategy. One possible way is to leverage independent relations in the measure of time series data, if any. For instance, if processes $X_1 := \{x_{t,1} | t \in T\}$ and $X_2 := \{x_{t,2} | t \in T\}$ happen to be independent of processes $X_3$ and $X_4$, then we can just learn the underlying processes for $(X_1, X_2)$ and $(X_3, X_4)$ separately. Another potential way is to use the conditional independent relations in the measured time series data. For example, if processes $X_1$ and $X_2$ are independent from $X_3$ and $X_4$ given $X_5$ and $X_6$, then we can just learn the underlying processes for $(X_1, X_2, X_5, X_6)$ and $(X_3, X_4, X_5, X_6)$ separately. In this way, we can reduce the search space and further reduce the complexity even in high-dimensional time series data. We hope that some other developments in reducing the computational load in deep learning can also be helpful.

### H.2 NOISY ENVIRONMENTS

In some real-world scenarios, it is possible for the mixing process not to be invertible, for example, if the mixing process is highly noisy in each observed process, which might be the case in financial data (Hu & Schennach, 2008). There are some developments relying on the additive noise assumptions. If one makes strong assumptions on the noise, such as assuming an additive noise model, it is possible to develop a certain type of identifiability just like the extension of nonlinear ICA to the additive noise model case in (Khemakhem et al., 2020a). However, general approaches to deal with non-parametric noise terms to developed in this field in the future. To further evaluate the insight of this potential solution, we have conducted experiments on synthetic data. Specifically, we first follow Equation (2) to generate latent variables with temporal and instantaneous dependencies and generate observed variables by $\mathbf{x}_t = \mathbf{g}(\mathbf{z}_t, \varepsilon)$, where $\varepsilon$ is the noise introduced to the mixing process. Experiment results in Table A20 show that our method can still achieve relatively good identifiability results even in a noisy environment, proving the potential of our insight.

---

[3]http://mocap.cs.cmu.edu/

Table A20: The mean and standard deviation of MCC of the IDOL model under noise environment with different noise scales. The noise scale is defined as the ratio of the noise variance to the observation variance, expressed as a percentage.

| Noise Scale | 0.1 | 0.3 | 0.5 | 0.7 |
|---|---|---|---|---|
| MCC | 0.9257 | 0.8362 | 0.7381 | 0.6095 |

Table A21: The mean and standard deviation of MCC on subsets of dataset A with different sizes.

| Sample Size | 1,000 | 10,000 | 25000 | 50,000 | 100,000 |
|---|---|---|---|---|---|
| MCC | 0.853(0.102) | 0.884(0.011) | 0.912(0.025) | 0.945(0.023) | 0.965(0.029) |

### H.3 ANALYSIS OF SAMPLE SIZE

To assess the impact of sample size on identification performance, we conduct an ablation study focusing on sample size. Specifically, we create subsets of Dataset A containing 1,000, 10,000, 25,000, and 50,000 samples. For each dataset, we used the same hyperparameters, such as learning rate, random seed, and batch size. We repeated the experiment three times with different random seeds and reported the values of mean and standard deviation. The results in Table A21 show that as the sample size decreases, the performance of the model gradually declines. However, even with just 1k samples, our method achieves relatively good performance, demonstrating its robustness in small sample scenarios.

### H.4 SUBSAMPLED TIME SERIES

Sequentially, we consider another case where the time series data are sampled with low resolutions. When the time series data is sampled with low frequency, additional edges are introduced into the Markov network, making it denser. We can further assume a sparse mixing process Zheng et al. (2022) to achieve identifiability. Specifically, When conditioned on historical information that provides sufficient changes, the sparse mixing procedure assumption imposes structural constraints on the mapping from estimated to true latent variables. This compensates for potentially insufficient distribution changes, enabling identifiability even when the time series data with low-sample regimes. To further evaluate this insight, we also conduct experiments on synthetic datasets with low-sample regimes. Specifically, we first generate time series data with temporal and instantaneous dependencies. Then we randomly subsample the synthetic time series data. To enforce the sparsity of the estimated mixing process, we add an extra L1 constraint regarding the partial differential between the estimated observed and latent variables, i.e. $|\frac{\partial \hat{\mathbf{x}}_t}{\partial \hat{\mathbf{z}}_t}|_1$. Experiment results are shown in Table A22.

Table A22: The mean and standard deviation of MCC between the standard IDOL and the IDOL with sparse mixing constraint.

| Model | IDOL+Sparse Mixing Constraint | IDOL |
|---|---|---|
| MCC | 0.837(0.078) | 0.786(0.085) |

From the experimental results, we can draw the following conclusions: 1) When time series data is sampled at low frequency, the identifiable performance of the IDOL model decreases. This is because the causal process between latent variables becomes dense, which is consistent with our theoretical results. 2) by employing a sparse mixing constraint, we found that the identifiability performance of the model has been significantly improved, which proves that the above insight is effective.

