# OpenReview forum: "On the Identification of Temporal Causal Representation with Instantaneous Dependence"
_ICLR.cc/2025/Conference — ICLR 2025 Oral_

### Official Review · Reviewer_uLFc · 2024-10-30

**Soundness:** 3
**Presentation:** 3
**Contribution:** 3
**Rating:** 8
**Confidence:** 3

**Summary:**

This paper introduces a framework called IDOL (Identification framework for Instantaneous Latent dynamics) to enhance temporally causal representation learning for time series data with instantaneous dependencies. Traditional approaches for identifying latent causal processes in time series data often assume that the latent causal variables lack instantaneous interactions, limiting real-world applicability. IDOL addresses this limitation by applying a sparsity constraint on causal influences, allowing for both time-delayed and instantaneous dependencies in latent variables. The IDOL frameworkassumes a sparse influence within latent causal processes, allowing both time-delayed and instantaneous relations. Unlike prior methods that require data interventions or predefined groupings to achieve identifiability, IDOL elies on this sparse latent structure alone, making it highly applicable to scenarios where interventions are impractical. The framework’s theoretical foundation is built on leveraging sufficient variability and temporal contextual information, establishing identifiability through a combination of variational inference and sparsity regularization. This enables the model to accurately reconstruct latent variables and the underlying causal relationships without complex external assumptions.

**Strengths:**

The proposed IDOL framework moves beyond traditional methods that often rely on grouping of variables or direct interventions, by introducing a sparse influence assumption to capture the natural sparsity in many real-world datasets. This approach is novel in handling instantaneous dependencies without requiring interventions or grouping. Furthermore, the paper demonstrates rigorous theoretical and empirical quality, supported by a well-founded identifiability proof and a solid mathematical framework. Experimental validation on both synthetic and real-world human motion datasets further underscores the robustness and reliability of the model, showcasing its ability to accurately identify causal relationships and achieve high predictive accuracy is synthetic and real-world datasets. The paper is overall clearly written and easy to follow. Overall, this work is significant for the field, since causal discovery for time series with instantaneous effects is an important open problem.

**Weaknesses:**

The model assumes an invertible mixing process to reconstruct latent causal structures, which may not always be feasible in real-world data. In some scenarios, particularly in non-linear and noisy environments, this assumption could lead to inaccurate or incomplete latent representations, potentially undermining the model’s performance and causal interpretability. Furthermore, IDOL’s effectiveness heavily depends on the assumption of a sparse latent process. In cases where this sparsity assumption does not hold (i.e., when the causal structure is dense or complex), IDOL’s performance degrades, as demonstrated in the experiments. This sensitivity suggests that the framework may be less robust in scenarios where latent processes are highly interconnected.

**Questions:**

Can you please comment on the performance of your method in noisy environments and low-sample regimes?

---

> ### Author Response · Authors · 2024-11-20
> **Response to Reviewer uLFc, Part 1**
>
> Dear Reviewer uLFc, thank you for taking the time and effort to review our paper. Your valuable suggestions have been instrumental in helping us improve the scalability of our work, i.e., extending its applicability to noisy environments and low-sample regimes. We deeply appreciate your insightful feedback and thoughtful comments.
>
> >W1 (Q1): The model assumes an invertible mixing process to reconstruct latent causal structures, which may not always be feasible in real-world data. In some scenarios, particularly in non-linear and noisy environments, this assumption could lead to inaccurate or incomplete latent representations, potentially undermining the model’s performance and causal interpretability. Can you please comment on the performance of your method in noisy environments?
>
> **A1**: That is a great point! Thank you for this insightful suggestion! Indeed, if the mixing process is very noisy in each measured observed variable, for instance, in financial data, our model assumption is violated [1]. There are some developments that rely on the additive noise assumptions. If one makes strong assumptions on the noise, says, by assuming an additive noise model, then it is possible to develop a certain type of identifiability, just like the extension of nonlinear ICA to the additive noise model case in [2]. However, general approaches to dealing with non-parametric noise terms are still to be developed in this field in the future. In light of your question, we have conducted experiments on the synthetic datasets with different noise scales (scale=0.1 means that the variance of the noise is 0.1 times the variance of the observation), which are shown as follows.
>
>
> | Noise Scale |   0.0  |   0.1  |   0.3  |   0.5  |   0.7  |
> |:-----------:|:------:|:------:|:------:|:------:|:------:|
> |     MCC     | 0.9645 | 0.9257 | 0.8362 | 0.7381 | 0.6095 |
>
>
> The experimental results show that our method can still achieve relatively good identifiability results in a noisy environment of a reasonable noise scale. We have added this discussion in Appendix G.2.
>
> [1] Hu, Yingyao, and Susanne M. Schennach. "Instrumental variable treatment of nonclassical measurement error models." Econometrica 76.1 (2008): 195-216.
>
> [2] Khemakhem, Ilyes, et al. "Variational autoencoders and nonlinear ica: A unifying framework." International conference on artificial intelligence and statistics. PMLR, 2020.
>
> >W3: Furthermore, IDOL’s effectiveness heavily depends on the assumption of a sparse latent process. In cases where this sparsity assumption does not hold (i.e., when the causal structure is dense or complex), IDOL’s performance degrades, as demonstrated in the experiments. This sensitivity suggests that the framework may be less robust in scenarios where latent processes are highly interconnected.
>
>
>
> **A3**: Thank you for the good question! When the causal structure is dense, the identification degree depends on how the graph structure is dense. Let us consider the extreme situation: if there is only time-delay causal inference, which is very dense, and there are no instantaneous relationships, then we can still recover all the latent processes and the whole graph according to previous results [3]. If the instantaneous relationships are also dense, the problem becomes much more complex. It depends on the couple relationships of the time-delay and the instantaneous relations, as discussed at the end of Section 3.4. Specifically, for the latent variables, whose intimate neighbor set is an empty set, then those latent variables can be component-wise identifiable. For the latent variables, whose intimate neighbor set is not empty, each true variable can be a function of, at most, an estimated version of its corresponding variable and those within the intimate set.
>
> We hope that our responses answer to your questions.
>
> [3] Yao, Weiran, Guangyi Chen, and Kun Zhang. "Temporally disentangled representation learning." Advances in Neural Information Processing Systems 35 (2022): 26492-26503.

---

> ### Author Response · Authors · 2024-11-20
> **Response to Reviewer uLFc, Part 2**
>
> >W4 (Q2): Can you please comment on the performance of your method in low-sample regimes?
>
> **A4**: Thanks a lot for your question! We are not sure if you mean low-sample resolutions of the time series data or small sample size by low-sample regimes. To better address your concerns, we have considered these two cases as follows:
>
> - **Small sample size**: We first consider the case with a small sample size by low-sample regimes. In light of your questions, we have evaluated how the sample size influences the identification performance, and we have conducted experiments on synthetic data. Specifically, we create subsets of Dataset A containing 1,000, 10,000, 25,000, and 50,000 samples. For each dataset, we use the same hyperparameters, such as learning rate, random seed, and batch size. We repeated the experiment three times with different random seeds and reported mean and variance. As shown in the following table, as the sample size decreases, the performance of the model gradually decreases. However, even with 1,000 samples, our method can still achieve relatively good performance, which proves that our method has good robustness even in low-sample data sets. In light of your question, we have added experiment results and discussion to Appendix G.3.
>
>
> | Sample Size      |     1000     |     10000    |     25000    |     50000    |
> |:-----------:|:------------:|:------------:|:------------:|:------------:|
> |     MCC      | 0.853(0.102) | 0.884(0.011) | 0.912(0.025) | 0.945(0.023) |
>
>
>
> - **Low resolution**: Sequentially, we consider another case where the time series data are sampled with low resolutions [4], where additional edges are introduced into the Markov network, making it denser. In this case, our performance will drop as the identifiability becomes harder. We can further assume a sparse mixing process to achieve identifiability. Specifically, when conditioned on historical information that provides sufficient changes, the sparse mixing procedure assumption imposes structural constraints on the mapping from estimated to true latent variables. This compensates for potentially insufficient distribution changes, enabling identifiability even when the time series data has low-sample regimes. To further evaluate this insight, we also conduct experiments on a synthetic downsampled dataset with a sparse mixing function. As shown in the following Table, when we constrain the sparsity of the mixing process, our method can still achieve good performance. In light of your question, we have added this discussion to Appendix G.4.
>
> | Model | IDOL+Sparse Mixing Constraint |     IDOL     |
> |:-----:|:-----------------------------:|:------------:|
> |  MCC  |          0.837(0.078)         | 0.786(0.085) |
>
> [4] Danks, D. and Plis, S. Learning causal structure from undersampled time series. In JMLR: Workshop and Conference Proceedings, 2014.

---

> ### Author Response · Authors · 2024-11-22
> **Could you please let us know whether our responses and updated submission properly addressed your concerns?**
>
> Dear Reviewer uLFc,
>
> We would like to express our sincere gratitude for your time to review our manuscript. As the rebuttal discussion period is coming to an end, we are eagerly anticipating any additional feedback you may have. We are thrilled to engage in further discussions with you.
>
> Best regards,
>
> Authors of submission 4912

---

> ### Author Response · Authors · 2024-11-25
> **Could you please let us know whether our responses and updated submission properly addressed your concerns?**
>
> Dear Reviewer uLFc,
>
> Thank you for your valuable time in reviewing our submission and for your insightful suggestions to make experiments more complete. We've tried our best to conduct the experiments and address your concerns in the response and updated submission. Due to the limited rebuttal discussion period, we eagerly await any feedback you may have regarding these changes. If you have further comments, please kindly let us know--we hope for the possible opportunity to respond to them.
>
> Many thanks,
>
> Authors of submission #4912

---

> > ### Comment · Reviewer_uLFc · 2024-11-26
> >
> > Thank you for your reply. After reading your reply as well as the other reviewers comments, I decided to raise my score accordingly.

---

> ### Author Response · Authors · 2024-11-26
>
> Dear Reviewer uLFc,
>
> We are glad to address your concerns. In light of your treasurable suggestions, we have provided more discussions and experiments about the applicability to noisy environments and low-sample regimes.
>
> With best wishes,
>
> Authors of submission #4912

---

### Official Review · Reviewer_QmDF · 2024-11-01

**Soundness:** 3
**Presentation:** 3
**Contribution:** 3
**Rating:** 8
**Confidence:** 2

**Summary:**

The paper presents a framework, IDOL (identification of instantaneous Latent Dynamics), for identifying temporally causal representation with instantaneous dependencies. IDOL employs a sparse latent process assumption, which is more adaptable to real-world data. The framework is validation through extensive experiments on both synthetic and rea-world human motion datasets.

**Strengths:**

The paper introduces a novel approach to identifying temporally causal representations in time series data with instantaneous dependencies. This approach addresses a gap by proposing a sparse latent process assumption that is more practical for real-world applications than previous assumptions.

Extensive evaluations are performance to demonstrate the effectiveness of the proposed approach.

The paper is well-organized. The use of illustrative figures helps clarify the complex concepts.

**Weaknesses:**

Providing further discussions on the possibility of extending IDOL to handle high-dimensional data can be beneficial.

Given the limitation due to the dependency on invertible mixing processes, providing guidelines for real-world applicability would add value.

**Questions:**

How does IDOL handle cases where the latent process sparsity assumption is only partially met?

Could the authors clarify the computational complexity of IDOL compared to baselines, especially for high-dimensional data?

Are there specific real-world scenarios where IDOL might struggle due to non-invertible mixing processes?

---

> ### Author Response · Authors · 2024-11-20
> **Response to Reviewer QmDF, Part 1**
>
> Dear Reviewer QmDF, we are very grateful for your valuable comments, helpful suggestions, and encouragement. Your insights into the further application of our IDOL model in real-world scenarios have greatly helped us bridge the gap between theory and practice. We provide the point-to-point response to your comments below and have updated the paper accordingly.
>
> >W1: Providing further discussions on the possibility of extending IDOL to handle high-dimensional data can be beneficial.
>
> **A1**: Thank you for your suggestion. We would like to kindly emphasize that our theorem is applicable to scenarios of any dimensionality despite possible performance loss in high-dimensional problems owing to the complexity of causal structure [1,2].
>
> To better address this challenge, we have proposed several potential solutions to more effectively address the challenges of high-dimensional time-series data. One idea is to make use of the divide-and-conquer strategy. One possible way is to leverage independent relations in the measure of time series data, if any. For instance, if processes $X _1:=$ { $x _{t,1} ∣t\in T$} and $X _2:=$ { $x _{t,2} ∣t\in T$} happen to be independent of processes $X _3$ and $X _4$, then we can just learn the underlying processes for $(X _1, X _2)$ and $(X _3, X _4)$ separately. Another potential way is to use the conditional independent relations in the measured time series data. For example, if processes $X _1$ and $X _2$ are independent from $X _3$ and $X _4$ given $X _5$ and $X _6$, then we can just learn the underlying processes for $(X _1, X _2, X _5, X _6)$ and $(X _3, X _4, X _5, X _6)$ separately. In this way, we can reduce the search space and further reduce the complexity even in high-dimensional time series data.
>
> In light of your constructive suggestion, we have added this discussion to Appendix G.1.3, and a unified strategy is to be developed.
>
> [1] Lopez, Romain, et al. "Large-scale differentiable causal discovery of factor graphs." Advances in Neural Information Processing Systems 35 (2022).
> [2]Cheng, Yuxiao, et al. "CUTS+: High-dimensional causal discovery from irregular time-series." Proceedings of the AAAI Conference on Artificial Intelligence. Vol. 38. No. 10. 2024.
>
> >W2 and W5(Q3): Given the limitation due to the dependency on invertible mixing processes, providing guidelines for real-world applicability would add value. and  Are there specific real-world scenarios where IDOL might struggle due to non-invertible mixing processes?
>
>
> **A2**: Thanks for your valuable suggestion. Indeed, in certain scenarios, it is possible for the mixing process not to be invertible, for instance, if the mixing process is highly noisy in each observed process, which might be the case in financial data [3]. We hope the community is able to address this problem in the near future. However, even if the observed processes seem highly dependent or even redundant, the mixing processes can still be invertible. Consider the following example. Suppose the underlying hidden processes are $z _1$ and $z _2$ and assume we observe three processes $x _1 = z _1+z _2, x _2 = z _1 - z _2$, and $x _3 = z _1 + 0.5 \times z _2$. Then, after preprocessing the observed processes with dimension reduction, we have two transformed observed processes, and the mixing procedure becomes square (with the same number of latent processes and observed processes) and invertible.
>
> [3] Hu, Yingyao. "The econometrics of unobservables: Applications of measurement error models in empirical industrial organization and labor economics." Journal of econometrics 200.2 (2017): 154-168.
>
> >W3(Q1): How does IDOL handle cases where the latent process sparsity assumption is only partially met?
>
> **A3**: Thank you for raising this important point. We hope you will find the discussion at the end of Section 3.4 helpful. Specifically, even if the sparsity assumption is only partially met, "each true variable can still be a function of, at most, an estimated version of its corresponding variable and those within the intimate set.
>
> Depending on the proposes, identifiability up to the subspace level can be sufficient, because we may be able to consider them as parts of a single macro variable. Let us provide a simple example here. In a video of a moving car, it might be hard to have individual identifiability of the separate car wheels and car body; however, they can be considered as essential parts of the macro variable 'car'. This macro representation might be sufficient for the purpose of modeling the interactions between the car and other objects.
>
> In light of your question, we have added the example of a moving car video to the end of Section 3.4 to improve our readability.

---

> ### Author Response · Authors · 2024-11-20
> **Response to Reviewer QmDF, Part 2**
>
> >W4(Q2): Could the authors clarify the computational complexity of IDOL compared to baselines, especially for high-dimensional data?
>
> **A4**: We appreciate this question and the opportunity to elaborate on IDOL's computational complexity in comparison to baseline methods, especially in the context of high-dimensional data. Specifically, we have provided a model efficiency analysis for the proposed IDOL and the baseline methods in Appendix F of the revised manuscript.
>
> Specifically, we have provided model efficiency comparison on the low-dimension dataset (e.g., the Human dataset) and the high-dimension dataset (the MoCap dataset) by evaluating the model efficiency from three aspects: forecasting performance, training speed, and memory footprint. We found that in low-dimensional datasets, IDOL performs nearly as efficiently as top methods like Autoformer and MICN and outperforms others like CARD. Meanwhile, in the dataset with 117 observations, whose dimension is much higher, the proposed IDOL method requires more training time due to the added complexity of Jacobian calculations for instantaneous effects. When dealing with high-dimensional datasets, such as pixel-level images or videos, where the dimensionality is high yet often redundant, a common strategy is to first utilize pre-trained low-dimensional representations of the pixels.

---

> ### Author Response · Authors · 2024-11-25
> **Have the concerns been adequately addressed in the response and revision?**
>
> Dear Reviewer QmDF,
>
> Thank you for dedicating time to review and provide feedback on our submission. We hope our response and revised work effectively address your concerns. If there are additional matters you'd like us to consider, we eagerly await the opportunity to respond.
>
> Best regards,
>
> Authors of submission #4912

---

### Official Review · Reviewer_mQKF · 2024-11-10

**Soundness:** 3
**Presentation:** 3
**Contribution:** 2
**Rating:** 8
**Confidence:** 4

**Summary:**

This paper proposes IDOL, a framework for achieving identifiability in sequential latent variable models with instantaneous dependencies. The authors establish identifiability up to permutation of the latent variables and demonstrate that the underlying causal graph can be identified up to its Markov equivalence class (if this interpretation is correct). They thoroughly discuss the limitations of their assumptions in comparison to recent works, which helps underscore the significance of the proposed framework.

An estimation method is also introduced, with experiments on synthetic data verifying the theoretical results, while real-world experiments highlight the importance of incorporating instantaneous dependencies.

Edit: All the major concerns have been addressed in the rebuttal, and hence I have raised my score to 8.

**Strengths:**

- The manuscript is clear in terms of motivating the problem and introducing the theoretical framework.
- Incorporation of instantaneous effects into sequential latent variable models is a very significant contribution.
- The paper discusses limitations of the assumptions in comparison to recent works.
- The experiments with real-world data motivate the incorporation of instantaneous effects.

**Weaknesses:**

**Minor Concerns**
- **Computational Complexity:** The sparsity constraint introduced in Eq. (11) seems to introduce significant computational complexity to the algorithm. The paper would benefit from a more detailed analysis regarding this. For example, would it be possible to compute wall-clock times (in training) for IDOL in comparison the proposed baselines?
- **Scalability to High-Dimensional Data:** The authors acknowledge limitations with respect to high-dimensional data, which can restrict the application to real-world scenarios. An experiment to understand how high-dimensional one can go with IDOL would be ideal to support your point to understand how high one could go for IDOL.

**Major Concern: Theory Section Clarity and Limitations**

The paper’s theoretical claims, particularly around identifiability, would benefit from clarification to avoid potential misunderstandings regarding the nature of identifiability achieved. It appears that IDOL identifies the latent Markov Network rather than the true causal graph for the instantaneous component of the latent dynamics. This is an important distinction, as conditional independence relations allow only for the identification of the Markov equivalence class, not the directed causal structure itself. However, the presentation throughout the paper, especially in the introduction, experiments (such as Figure 4), and conclusions, may lead readers to infer that IDOL identifies the causal graph rather than the Markov network.

To address this issue, the authors could consider the following changes:

- **Introduction (around line 89):** Indicate that the identifiability of the instantaneous structure in IDOL is only up to a Markov equivalence class, clarifying that IDOL does not identify the directions of edges in the instantaneous part.
- **Figure 1c Modification:** Consider modifying Figure 1c to remove the arrow pointers from edges, signaling that the result is a Markov network rather than a causal graph when discussing identifiability (this might make sense in terms of theory, but not from a data generation perspective).
- **Conclusion:** Mention the Markov equivalence class limitation explicitly. This would open a path for further research to extend the identifiability result from Markov equivalence to the full causal structure, especially given the promising empirical results observed in Figure 4.

The following specific statements in the theory section could be revised to improve clarity and accuracy:

- lines 130-132: “the latent causal relations are also immediately identifiable because conditional independence relations fully characterize instantaneous causal relations in an instantaneous causally sufficient system”. I don’t think this line is correct without any additional assumptions. Conditional independence relations only provide the Markov equivalence class, not the exact causal graph, without further assumptions. Rephrasing this to accurately reflect the distinction between the Markov equivalence class and the causal graph would strengthen the theoretical foundation.
- lines 171-172: Could you indicate whether $p_{c_t}$ refers to the marginal distribution $p(c_t)$ or the conditional distribution $p(c_t|z_{t-2})$?
lines 165-188: For better readability, could you indicate $c_t \in R^{2n}$ in your example? Otherwise, at first glance it reads as  $\{z_{t,i}, z_{t-1,i} \}$ for $c_{t,i}$ in Theorem 1.
- line 217: Would it be better to use $\emptyset$ to refer to $\Phi$ as an empty set?
- line 230: Could you define “isomorphic” for Markov networks? A footnote or reference to the Appendix suffices.

**Questions:**

- line 41: do you mean “mixing function” instead of “mixture function”?
- lines 332, 386 and 387: Notation. You are using $\mathcal{L}$ and $L$ interchangeably. Could you revise this?
- If I am not mistaken, your identifiability theory does not obtain the causal graph, but a markov equivalence of it (please correct if mistaken). Yet apparently, the synthetic experiments suggest that you estimate the instantaneous causal graph with 100% accuracy (Figure 4, bottom left). Could you provide some explanation for this? For example, is it possible that your assumptions allow for stronger identifiability results that are overlooked in the presented theory?

---

> ### Author Response · Authors · 2024-11-20
> **Response to Reviewer mQKF, Part 1**
>
> Dear Reviewer mQKF, thank you for your insightful and constructive feedback. Your comments have significantly helped us refine the rigor and completeness of our theoretical analysis. Additionally, your suggestions have guided us to improve the clarity and readability of our writing, making our work more accessible to a broader audience. We greatly appreciate the time and effort you dedicated to reviewing our submission.
>
> >W1.1: Computational Complexity: The sparsity constraint introduced in Eq. (11) seems to introduce significant computational complexity to the algorithm. The paper would benefit from a more detailed analysis regarding this. For example, would it be possible to compute wall-clock times (in training) for IDOL in comparison the proposed baselines?
>
> **A1.1**: Thank you for this valuable suggestion. We appreciate your focus on the practicality of training time, which is indeed a crucial factor for the usability of our approach. In response to your feedback, we have included the wall-clock training times for the proposed IDOL method and the baseline methods in Appendix F.1 of the revised version.
>
> For the wall-clock training times of different methods, we used a consistent hardware setup, including the same GPU, CPU, and memory configurations, to ensure comparability. To measure the actual time for training, we wrapped the training process in a timer within our code. The Python pseudo-code is provided below:
> ```python
>     import time
>     start_time = time.time()
>     model.train(training_data)  # Model training process
>     end_time = time.time()
>     wall_clock_time = end_time - start_time  # Calculate wall-clock time in seconds
> ```
> To ensure reliability, we run these codes three times and average the results, which helps to smooth out any minor variations due to random factors in the environment. The wall-clock training of the proposed IDOL model and other baselines on the Walking dataset are shown in the following table.
>
> | Methods | IDOL           | TDRL          | CARD          | FITS          | MICN           | iTransformer   | TimesNet        | Autoformer     |
> |---------|----------------|---------------|---------------|---------------|----------------|----------------|-----------------|----------------|
> | Second  | 65.960(7.556)   | 33.902(5.781) | 62.994(3.035) | 92.941(0.715) | 45.547(13.034) | 38.155(6.441)  | 324.941(25.286) | 51.648(15.673) |
>
>
> According to the experiment results,
>
> - Compared to our baseline model TDRL, the wall-clock training time of the proposed IDOL is nearly twice as slow. Theoretically speaking, the primary computational difference is that IDOL calculates the Jacobian matrix for both time-delayed and instantaneous relationships, while TDRL only calculates the time-delayed component.
> - Compared to other mainstream baselines, our IDOL method is slower than some models like MICN and iTransformer. However, our method is still faster than models like FITS.

---

> ### Author Response · Authors · 2024-11-20
> **Response to Reviewer mQKF, Part 2**
>
> >W1.2: Scalability to High-Dimensional Data: The authors acknowledge limitations with respect to high-dimensional data, which can restrict the application to real-world scenarios. An experiment to understand how high-dimensional one can go with IDOL would be ideal to support your point to understand how high one could go for IDOL.
>
> **A1.2**: We sincerely appreciate the insightful comment about the importance of evaluating the scalability of our approach with respect to high-dimensional data. Although the difficulty of causal process identification increases with dimensionality [1,2], some identification can still be achieved. In light of your suggestions, we have conducted experiments on the simulation and real-world datasets with varying dimensionality for a better understanding of the limits of IDOL’s performance in high-dimensional scenarios.
>
> As for the experiment on the synthetic datasets, we follow the same data generation process to generate simulation datasets with latent variable dimensions of 8, 16, 24, and 32. All these datasets share a similar latent causal process: chain-like instantaneous effects and ono-to-one temporal effects. We then measured the Mean Correlation Coefficient (MCC) between the ground truth $\mathbb{z} _t$ and the estimated $\hat{\mathbb{z}}_t$. The experimental results are presented in the following table.
>
> | Dimension |   8   |   16   |    24    |   32   |
> |:---------:|:------:|:------:|:-------:|:------:|
> |    MCC    | 0.9801(0.002) | 0.9747 (0.0029) | 0.9243 (0.0173) | 0.8640 (0.0071) |
>
> According to the results, as the dimension of latent variables increases, the value of MCC is still acceptable, despite possible performance loss in high-dimensional problems.
>
> Besides, we have also conducted experiments on a high-dimensional real-world dataset CMU-MoCap. The dataset contains various motion capture recordings and 117 skeleton-based measurements. Here are the results.
>
> |  Motion | Predicted  Length |  IDOL |       |  TDRL |       |  CARD |       |  FITS |       |  MICN |       | iTransformer |       | TimesNet |       | Autoformer |       |
> |:-------:|:-----------------:|:-----:|:-----:|:-----:|:-----:|:-----:|:-----:|:-----:|:-----:|:-----:|:-----:|:------------:|:-----:|:--------:|:-----:|:----------:|:-----:|
> |         |                   |  MSE  |  MAE  |  MSE  |  MAE  |  MSE  |  MAE  |  MSE  |  MAE  |  MSE  |  MAE  |      MSE     |  MAE  |    MSE   |  MAE  |     MSE    |  MAE  |
> | Running |       50-25       |  0.110 | 0.082 | 0.448 | 0.108 | 0.998 | 0.135 | **0.076** | **0.064** | 0.658 | 0.135 |     0.458    | 0.106 |   2.616  | 0.202 |    1.033   | 0.179 |
> |         |       50-50       | 0.286 | 0.109 | 1.779 | 0.167 | 2.217 | 0.153 | **0.266** | **0.103** | 0.978 | 0.161 |     1.831    |  0.170 |   4.720   | 0.252 |    3.944   | 0.283 |
> |  Soccer |       50-25       | **0.022** | **0.043** | 0.026 | 0.047 | 0.206 | 0.082 | 0.076 | 0.065 | 0.057 | 0.066 |     0.032    | 0.051 |   0.063  | 0.068 |    0.211   | 0.105 |
> |         |       50-50       | **0.079** | **0.071** | 0.084 | 0.073 | 0.397 | 0.108 | 0.265 | 0.103 | 0.284 |  0.120 |     0.133    | 0.082 |   0.392  | 0.107 |    0.452   | 0.143 |
>
> As shown in the table above, the IDOL model achieved a comparable forecasting performance in the high-dimensional dataset Running.
>
> In addition, we further provided some potential solutions to handle high-dimensional scenarios. One idea is to make use of the divide-and-conquer strategy. One possible way is to leverage independent relations in the measure of time series data, if any. For instance, if processes $X _1:=${$x _{t,1} ∣t\in T$} and $X _2:=${$x _{t,2} ∣t\in T$} happen to be independent of processes $X _3$ and $X _4$, then we can just learn the underlying processes for $(X _1, X _2)$ and $(X _3, X _4)$ separately. Another potential way is to use the conditional independent relations in the measured time series data. For example, if processes $X _1$ and $X _2$ are independent from $X _3$ and $X _4$ given $X _5$ and $X _6$, then we can just learn the underlying processes for $(X _1, X _2, X _5, X _6)$ and $(X _3, X _4, X _5, X _6)$ separately. In this way, we can reduce the search space and further reduce the complexity even in high-dimensional time series data. We hope that some other developments in reducing the computational load in deep learning can also be helpful.
>
> Thank you again for your valuable suggestions. We have added this discussion to Appendix G to enhance the integrity of our paper.
>
> [1] Lopez, Romain, et al. "Large-scale differentiable causal discovery of factor graphs." Advances in Neural Information Processing Systems 35 (2022).
> [2] Cheng, Yuxiao, et al. "CUTS+: High-dimensional causal discovery from irregular time-series." Proceedings of the AAAI Conference on Artificial Intelligence. Vol. 38. No. 10. 2024.

---

> ### Author Response · Authors · 2024-11-20
> **Response to Reviewer mQKF, Part 3**
>
> >W2.1: **Major Concern: Theory Section Clarity and Limitations**: The paper’s theoretical claims, particularly around identifiability, would benefit from clarification to avoid potential misunderstandings regarding the nature of identifiability achieved. It appears that IDOL identifies the latent Markov Network rather than the true causal graph for the instantaneous component of the latent dynamics. This is an important distinction, as conditional independence relations allow only for the identification of the Markov equivalence class, not the directed causal structure itself. However, the presentation throughout the paper, especially in the introduction, experiments (such as Figure 4), and conclusions, may lead readers to infer that IDOL identifies the causal graph rather than the Markov network.
>
>
> >To address this issue, the authors could consider the following changes:
>
> >Introduction (around line 89): Indicate that the identifiability of the instantaneous structure in IDOL is only up to a Markov equivalence class, clarifying that IDOL does not identify the directions of edges in the instantaneous part.
>
> >Figure 1c Modification: Consider modifying Figure 1c to remove the arrow pointers from edges, signaling that the result is a Markov network rather than a causal graph when discussing identifiability (this might make sense in terms of theory, but not from a data generation perspective).
>
> >Conclusion: Mention the Markov equivalence class limitation explicitly. This would open a path for further research to extend the identifiability result from Markov equivalence to the full causal structure, especially given the promising empirical results observed in Figure 4.
>
> **A2.1**: Thank you very much for your so insightful, helpful, and constructive reviews. To clarify, we mean that the latent causal relationships are (partially) identifiable. Due to the presence of temporal information, it is possible to go beyond the equivalence class. With certain additional assumptions, one can achieve full identifiability of the graph over the latent processes. Specifically, here are some take-home messages:
> - **Time-delayed edges**: All directions of time-delayed edges can be naturally determined, as the direction of time is known.
> - **Instantaneous edges**: For any pair of adjacent latent variables $z_{t,i}, z_{t,j}$ at time step $t$, if their time-delayed parents are not identical, i.e., $Pa _ d(z _{t,i})\not=Pa _ d(z _ {t,j})$, the direction of edge between them becomes identifiable.
>
> In light of your valuable comments, we have carefully followed your suggestions and made the necessary changes.
> - Line 21-24 in the abstract: '..., we establish identifiability results of the latent causal process up to a Markov equivalence class ... We further explore under what conditions the identification can be extended to the causal graph.'
> - Line 95-97 in the introduction: '..., which implies the identification of Markov equivalence class. Furthermore, we can extend to the identification of the causal graph when the endpoints of instantaneous edges do not share identical time-delayed parents.'
> - Line 54-63 in Figure 1(c), we did not modify the figure since in this case the causal graph is identifiable.
> - Line 528-531 in the conclusion: 'This paper proposes a general framework for time series data with instantaneous dependencies to identify the latent variables and latent causal relations up to the Markov equivalence class. Furthermore, with mild assumption, the causal graph is also identifiable.'
> - Appendix B.5: We have provided a detailed discussion about the conditions under which the causal graph is identifiable.
>
> Thank you once again for your valuable comments and suggestions, which have greatly helped clarify our theoretical results and contribute meaningfully to the field.
>
> >W2.2: lines 130-132: “the latent causal relations are also immediately identifiable because conditional independence relations fully characterize instantaneous causal relations in an instantaneous causally sufficient system”. I don’t think this line is correct without any additional assumptions. Conditional independence relations only provide the Markov equivalence class, not the exact causal graph, without further assumptions. Rephrasing this to accurately reflect the distinction between the Markov equivalence class and the causal graph would strengthen the theoretical foundation.
>
> **A2.2**: Thank you for your careful reading and correct comment. As discussed in Weakness 2.1, here we mean that 'the latent causal process is (partially) identifiable,' and an extra assumption is required to achieve full identifiability.
>
> To enhance the clarity and accuracy of our work, we have rephrased the statement in Lines 128–130 as "... up to a Markov equivalence class. We further show how to go beyond the Markov equivalence class and identify the instantaneous causal relations with a mild assumption in Corollary A2"

---

> ### Author Response · Authors · 2024-11-20
> **Response to Reviewer mQKF, Part 4**
>
> >W2.3: Lines 171-172: Could you indicate whether $p_{\mathbf{c} _t}$ refers to the marginal distribution $p(\mathbf{c} _t)$ or the conditional distribution $p(\mathbf{c} _t|\mathbf{z} _{t-2})$?
>
> **A2.3**: Thanks for the question! Here $p _{\mathbf{c} _t}$ refers to the conditional distribution $p(\mathbf{c} _t|\mathbf{z} _{t-2})$. We have changed $p _{\mathbf{c} _t}$ to $p(\mathbf{c} _t|\mathbf{z} _{t-2})$ in Theorem 1 and its counterpart in the appendix for better clarity.
>
> >W2.4: Lines 165-188: For better readability, could you indicate $\mathbf{c} _t\in \mathbb{R}^{2n}$ in your example? Otherwise, at first glance it reads as $z _{t,i}, z _{t-1,i}$ for $\mathbf{c} _{t,i}$ in Theorem 1.
>
> **A2.4**: We appreciate your efforts in helping us improve the readability. We have incorporated the changes in all theorems, for example, in lines 164, 180-183, 218, etc.
>
> >W2.5: line 217: Would it be better to use $\emptyset$ to refer to $\Phi$ as an empty set?
>
> **A2.5**: Thanks a lot! For better readability, we have updated it to $\emptyset$ in line 213 of the revised paper.
>
> >W2.6: line 230: Could you define “isomorphic” for Markov networks? A footnote or reference to the Appendix suffices.
>
> **A2.6**: We deeply value your careful reviews for the preciseness of our work. In light of your suggestion, we have defined "isomorphic" for Markov networks. Specifically, we let the $V(\cdot)$ be the vertex set of any graphs. An isomorphism of Markov networks $M$ and $\hat{M}$ is a bijection between the vertex sets of $M$ and $\hat{M}$
>
> $$f:V(M)\rightarrow V(\hat{M})$$
>
> such that any two vertices $u$ and $v$ of $M$ are adjacent in $G$ if and only if $f(u)$ and $f(v)$ are adjacent in $\hat{M}$.
>
> In light of your suggestions, we have added a footnote on page 5 and added this definition in Appendix B.7.
>
> >W3(Q1): line 41: do you mean “mixing function” instead of “mixture function”?
>
> **A3**: Thank you for your careful review, which has improved the clarity and consistency of our paper. You are correct that the 'mixture function' should be the 'mixing function.' In response to your suggestion, we have reviewed the entire paper and replaced the 'mixture function' with the 'mixing function' in line 42 to ensure consistency throughout the text.
>
> >W4(Q2): Lines 332, 386 and 387: Notation. You are using $\mathcal{L}$ and $L$ interchangeably. Could you revise this?
>
> **A4**: Thanks for your careful review. We have gone through the whole paper and use $\mathcal{L}$ to denote the objective loss. The modifications are made in lines 333 and 387.
>
> >W5(Q3): If I am not mistaken, your identifiability theory does not obtain the causal graph, but a markov equivalence of it (please correct if mistaken). Yet apparently, the synthetic experiments suggest that you estimate the instantaneous causal graph with 100% accuracy (Figure 4, bottom left). Could you provide some explanation for this? For example, is it possible that your assumptions allow for stronger identifiability results that are overlooked in the presented theory?
>
> **A5**: Thank you for this insightful question, which helps us improve the solidness of our experiment. As discussed in W2.1, the true causal graph is identifiable with mild assumption. The true causal graph of this experiment  (Figure 4) satisfies the assumption and is indeed fully identifiable.
>
> Due to the limitations of Markdown's expressiveness, we present the causal graph in tabular form as follows. A clearer version of the graph can be found in Figure A6 in line 1359 of the revised paper.
>
> | $z _{t-1,1}$     | $\rightarrow$ | $z _{t,1}$ |
> |-----------|---------|---------|
> | $\downarrow$   | $\nearrow$     | $\downarrow$     |
> | $z _{t-1,2}$   | $\rightarrow$     | $z _{t,2}$      |
> | $\downarrow$   |      | $\downarrow$     |
> | $z _{t-1,3}$   | $\rightarrow$     | $z _{t,3}$      |
>
> Since the skeleton and directions of time-delayed edges are straightforward to determine, we primarily focus on analyzing the directions of instantaneous edges within $\mathbb{z} _t$.
>
> Since $z _{t-1,3}\rightarrow z _{t,3}\leftarrow z _{t,2}$ is a v-structure, $z _{t,3}\leftarrow z _{t,2}$ can be determined. Since $z _{t-1,1}\rightarrow z _{t,1}\rightarrow z _{t,2}$ is a chain, $z _{t-1,1}$ and $z _{t,2}$ are not adjacent, and $z _{t-1,1}\rightarrow z _{t,1}$ is known, we have $z _{t,1}\rightarrow z _{t,2}$. Thus, the causal graph is identifiable.
>
> For better readability and clarity, we have added this discussion in Appendix B.5. We have also mentioned it in the experiment part (lines 460-462): 'Please note that here, not only the Markov equivalence class but also the causal graph can be identified for dataset A, as shown in Figure 4. Please refer to Appendix B.5 for more details.

---

> > ### Comment · Reviewer_mQKF · 2024-11-20
> > **Response to authors**
> >
> > Dear authors,
> >
> > Thank you very much for carefully addressing the concerns. After reading your answers and the updated paper, I can see that my concerns have been covered. I am very excited to see some theoretical results that allow to go beyond the Markov equivalence class on the instantaneous dependences. Hence, I am raising my score to 8.

---

> > > ### Author Response · Authors · 2024-11-20
> > >
> > > Dear Reviewer mQKF,
> > >
> > > We are delighted that you found the response well addressed your concerns. Thank you once again for your valuable comments and suggestions!
> > >
> > > With best wishes,
> > >
> > > Authors of submission #4912

---

### Meta-Review · Area_Chair_wRMJ · 2024-12-18

**Metareview:**

The authors propose a framework to identify temporally causal relations with instantaneous dependencies.   The three reviewers all voted to accept the paper, noting that the problem is important and motivated well and that the incorporation of instantaneous effects is a significant contribution. During the discussion phase, two reviewers raised their scores due to the satisfactory additional work presented to address their comments. The authors are encouraged to incorporate additional information in the final draft if space allows.

**Additional Comments On Reviewer Discussion:**

Two of the reviewers raised their scores in response to the additional information provided by the authors. All three voted Accept.

---

### Decision · Program_Chairs · 2025-01-22

Accept (Oral)